# Masked Vision and Language Modeling for Multi-modal Representation Learning

**Gukyeong Kwon, Zhaowei Cai, Avinash Ravichandran,**
**Erhan Bas, Rahul Bhotika, Stefano Soatto**
AWS AI Labs
{gukyeong,zhaoweic,soattos}@amazon.com

## Abstract

In this paper, we study how to use masked signal modeling in vision and language (V+L) representation learning. Instead of developing masked language modeling (MLM) and masked image modeling (MIM) independently, we propose to build joint masked vision and language modeling, where the masked signal of one modality is reconstructed with the help from another modality. This is motivated by the nature of image-text paired data that both of the image and the text convey almost the same information but in different formats. The masked signal reconstruction of one modality conditioned on another modality can also implicitly learn cross-modal alignment between language tokens and image patches. Our experiments on various V+L tasks show that the proposed method, along with common V+L alignment losses, achieves state-of-the-art performance in the regime of millions of pre-training data. Also, we outperforms the other competitors by a significant margin in limited data scenarios.

## 1 Introduction

Vision and language (V+L) representation learning has gained significant attention due to the transferablility of the representations to a diverse set of downstream tasks such as zero- or few-shot visual recognition (Jia et al., 2021; Radford et al., 2021; Tsimpoukelli et al., 2021), object detection (Cai et al., 2022; Kamath et al., 2021), information retrieval (Li et al., 2022; 2021), and multi-modal generation (Ramesh et al., 2022; 2021) etc. This success is mainly driven by large-scale pre-training with paired image and text data. The current V+L pre-training techniques particularly focus on the representation learning that characterizes the association between vision and language, and they are largely inspired by self-supervised learning techniques (Devlin et al., 2018; He et al., 2020) in uni-modal learning.

Masked signal modeling is a popular self-supervisory pre-training task (Devlin et al., 2018; Liu et al., 2019; Yang et al., 2019; Bao et al., 2021; Xie et al., 2021; He et al., 2022), which aims at reconstructing the masked signals from the unmasked ones. It has been independently explored in the domains of natural language processing (NLP) and computer vision (Devlin et al., 2018; Liu et al., 2019; Yang et al., 2019; Bao et al., 2021; Xie et al., 2021; He et al., 2022). For example, in the domain of NLP, BERT (Devlin et al., 2018) and several follow-up works (Liu et al., 2019; Yang et al., 2019) utilize masked language modeling (MLM) where the model is expected to predict the masked text tokens using unmasked tokens. They have shown that MLM leads to powerful generalization performance across diverse NLP tasks. In computer vision, as shown in Figure 1 (top-left), the masked image modeling (MIM) is to predict masked pixels or image patches using unmasked portions of the images. MIM has shown to be an effective pre-training task for learning visual representations (Bao et al., 2021; Xie et al., 2021; He et al., 2022).

While MLM and MIM have been actively explored in each domain, existing works do not fully leverage the masked multi-modal signal modeling in the domain of V+L. For example, as shown in Figure 1 (bottom-left), several approaches rely only on MLM with unmasked images and do not model the masked images (Duan et al., 2022; Li et al., 2022; 2021; 2019; Yang et al., 2022). In this case, the distribution of text given image, $p(T|I)$, can be learned, but the distribution of image given text, $P(I|T)$, cannot. This will potentially lead to biased performance in cross-modal retrieval tasks such as image-to-text or text-to-image retrieval as shown in our experiments. Although there

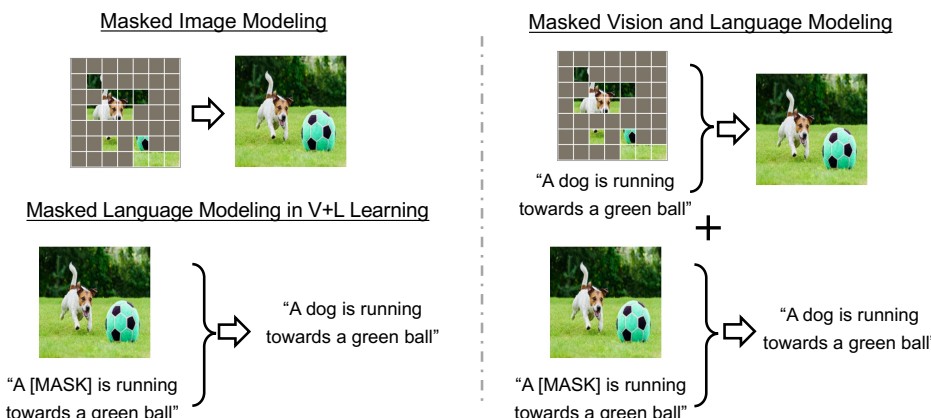

Figure 1: An overview of masked vision and language modeling. The left side shows existing approaches and the right side highlights our proposed approach.

exist works that use both modality signals masked, they either use a frozen object detector to extract region-based visual features (Chen et al., 2020b; Li et al., 2020a; Lu et al., 2020; Su et al., 2019; Tan & Bansal, 2019) or mask the image tokens from a pre-trained image tokenizer instead of the raw RGB pixels (Dou et al., 2022; Fu et al., 2021; Singh et al., 2022). These frozen object detector and image tokenizer not only require additional data for training but also prevent the V+L interactions from being learned end-to-end.

In this paper, we propose joint masked V+L modeling where the original signal is reconstructed by using its masked input and the corresponding unmasked input from the other modality. As illustrated in Figure 1 (right part), although we exploit random masking, the dog face in the image can be used to predict the masked text token "dog" and the text "green ball" can be used to reconstruct the corresponding masked patches in the image. To ensure that the model uses information from both modalities, we explicitly enforce the model to utilize cross-attention to generate the joint representations. Compared with the aforementioned existing works, our approach models both conditional distributions, $p(I|T)$ and $p(T|I)$. Also, the model is trained end-to-end, without frozen bottleneck components that disturb learning interactions between V+L. By reconstructing one modality signal from the corresponding the other modality signal (e.g. reconstructing the text "dog" from the visual signals of dog face), the model implicitly learns the alignment between V+L. In addition, we observe that the model trained for the joint masked V+L modeling becomes noticeably effective when the training data is limited. Overall, our contributions are summarized as below:

1. We propose a joint masked V+L modeling task. We show that models pre-trained with the proposed task, along with common V+L alignment tasks such as image-text matching, achieves state-of-the-art performance on a broad rage of V+L tasks.

2. We provide a probabilistic interpretation of the proposed method and highlight the difference between ours and existing approaches in terms of the V+L joint distribution estimation.

3. We achieve significantly better performance than other V+L models in the regimes of limited training data, and only ∼40% of data used by the state-of-the-art models is sufficient to match their performance.

## 2 RELATED WORK

**Vision and language representation learning** The methods in V+L representation learning can be categorized based on how the information is fused between the modalities to obtain the joint representations. We group the fusion techniques into three categories: 1) transformers with attention across modalities, 2) contrastive learning with a large-scale pre-training data, 3) a hybrid form of learning with cross-attention and a contrastive loss. The attention across modalities has been widely used with image features extracted from off-the-shelf object detectors and text features obtained from transformer encoders (Chen et al., 2020b; Li et al., 2020a; Lu et al., 2020; Tan & Bansal, 2019; Zhang et al., 2021; Li et al., 2020b; Su et al., 2019; Li et al., 2019). While cross-attention effectively aligns V+L representations, it is computationally expensive since all possible pairs of images

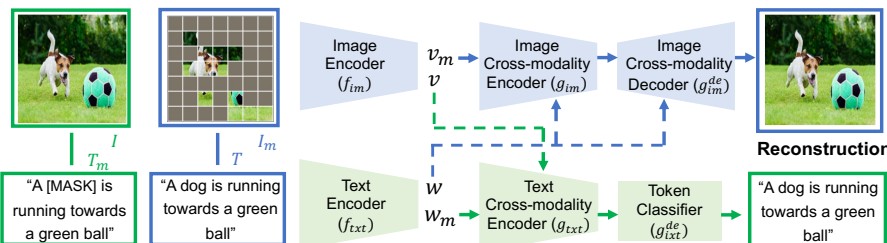

Figure 2: A framework of joint modeling of masked vision and language. The blue and green lines demonstrate the information flow for image and text reconstruction, respectively. The dotted lines indicate the cross-modal input of unmasked signals for generating attention.

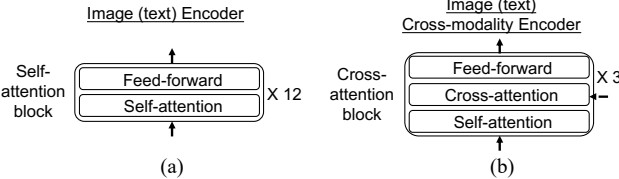

Figure 3: Visualization of image (text) encoders and image (text) cross-modality encoders.

and texts need to be processed. On the contrary, the authors in (Jia et al., 2021; Radford et al., 2021; Mokady et al., 2021; Shen et al., 2021; Yuan et al., 2021) show that contrastive learning with uni-modal encoders and millions of image-text pairs can achieve powerful zero-shot performance in diverse V+L tasks. The contrastive learning-based approaches do not rely on computationally expensive cross-attention but require an excessively large amount of training data. Hence, a combination of contrastive loss and cross-attention is utilized by complementing limitations of both approaches in (Li et al., 2021; 2022; Yang et al., 2022; Duan et al., 2022). In particular, only image and text pairs that result in high similarity by uni-modal encoders are processed using the cross-attention layers to reduce the computational burden and improve the alignment.

**Masked signal modeling** is a commonly used pre-training objective in the aforementioned V+L models. It has been independently explored in each of vision and language domain. In the NLP domain, BERT and its variants (Devlin et al., 2018; Liu et al., 2019) achieve representations that can generalize to a broad range of NLP tasks through MLM. Autoregressive language models (Radford et al., 2018; 2019) which predict masked future tokens have shown to be effective self-supervised learners. The success of the language models leads to several MIM techniques. BeiT (Bao et al., 2021) is trained to recover masked visual tokens which are obtained by a discrete variational autoencoder (dVAE). In SimMIM (Xie et al., 2021) and MAE (He et al., 2022), transformers are trained to recover masked patches in an end-to-end fashion. The authors in (Chen et al., 2020a) propose to autoregressively predict the unknown pixels to learn visual representations. In (Bachmann et al., 2022), multiple vision modality data are masked and reconstructed to learn visual representations. In the domain of V+L learning, (Arici et al., 2021) explores MIM and MLM for catalog data with short text attributes. V+L models with an object detector often aim at recovering only bounding box visual features (Chen et al., 2020b; Li et al., 2020a; Lu et al., 2020; Tan & Bansal, 2019; Su et al., 2019). Several V+L models focus on predicting future text tokens without MIM (Wang et al., 2021; Yu et al., 2022; Alayrac et al., 2022). While both MIM and MLM are explored in (Geng et al., 2022), the trained model is evaluated only on vision tasks. In (Dou et al., 2022; Fu et al., 2021; Singh et al., 2022; Wang et al., 2022), image tokens defined by image tokenizers trained with additional 250 million images (Ramesh et al., 2021) or distillation from the CLIP model (Radford et al., 2021) trained with 400 million image-text pairs (Peng et al., 2022) are reconstructed. In our work, we eliminate these model and data dependencies, by directly recovering RGB pixels and text tokens from masked image patches and masked text tokens. Therefore, MIM and MLM are seamlessly integrated to achieve generalizable V+L representations within a simple training framework.

## 3 METHOD

Our method has two types of pre-training objectives, which are 1) masked vision and language modeling and 2) multi-modal alignment. We explain each pre-training objective in this section.

## 3.1 Masked Vision and Language Modeling

The overall framework of masked vision and language modeling is shown in Figure 2. We use transformer-based encoders (Vaswani et al., 2017) for both image and text streams. Given an image-text pair $(I, T)$, an image encoder, $f_{im}$, is used to extract features, $\boldsymbol{v} = \{v_{cls}, v_1, ..., v_N\}$, from the image input $I$. $N$ is the number of image patches and $v_{cls}$ is the encoding of the image class token, [CLS]. The text encoder, $f_{txt}$, extracts features, $\boldsymbol{w} = \{w_{cls}, w_1, ..., w_M\}$, from the text input, $T$. $M$ is the number of text tokens and $w_{cls}$ is the encoding of the start token of a sentence, [START]. The image and the text encoder consist of 12 self-attention blocks as shown in Figure 3 (a). The image and the text features are further processed by image and text cross-modality encoders. The cross-modality encoders have 3 cross-attention blocks as illustrated in Figure 3 (b). The image (text) cross-modality encoder uses text (image) features to generate attentions. These cross-modality encoders can enhance the representation of one modality by interacting with another modality (Lu et al., 2020; Tan & Bansal, 2019).

**Image and Text Masking:** For text masking, we follow BERT (Devlin et al., 2018) with minor modifications. In BERT, the original tokens are replaced with either the [MASK] token or random tokens. We use only the [MASK] token to replace tokens to be masked (Wettig et al., 2022). For image masking, we follow (He et al., 2022; Xie et al., 2021) and use random masking of raw image patches with a masking patch size of $32 \times 32$. Given that $224 \times 224$ images are divided into $16 \times 16$ patches for the image encoder, the large masking patch prevents the model from simply copying their neighborhood pixels for reconstruction (Xie et al., 2021).

**Joint Reconstruction:** We reconstruct the original signals of one modality from its masked input conditioned on the unmasked input of the other modality. Specifically, an original image, $I$, and a masked text, $T_m$, are used to reconstruct an original text, $T$, and similarly a masked image, $I_m$, and an original text, $T$, are used to reconstruct the original image, $I$. For image reconstruction, $(I_m, T)$ is first given to the image and the text encoders to obtain masked image features, $\boldsymbol{v_m}$, and unmasked text features, $\boldsymbol{w}$. Following (Xie et al., 2021), we use both masked and unmasked patches to obtain $\boldsymbol{v_m}$. $(\boldsymbol{v_m}, \boldsymbol{w})$ are further processed by the image cross-modality encoder, $g_{im}$, where $\boldsymbol{w}$ is used to compute cross-attentions. The output of $g_{im}$ is mapped back to the original RGB image space by an image cross-modality decoder, $g_{im}^{de}$, which consist of 3 cross-attention blocks followed by a fully connected layer (FC). Although existing work exploits a light-weight transformer decoder with only self-attention (He et al., 2022) or a simple linear mapping (Xie et al., 2021) for the image decoder, we use joint information between modalities to allow further interactions in decoding. For masked text reconstruction, a token classifier, $g_{txt}^{de}$, which consists of a FC followed by softmax is applied to the output of the text cross-modality encoder, $g_{txt}$, for the token prediction. The masked V+L modeling loss, $\mathcal{L}_{MVLM}$, is defined as

$$\mathcal{L}_{MVLM} = \mathbb{E}_{(I,T) \sim D} \big[ \underbrace{\mathcal{H}(y_T^M, \phi_{txt}^M(I, T_m))}_{\text{MLM}} + \underbrace{\frac{1}{\Omega(I^M)} \| I^M - \phi_{im}^M(I_m, T) \|_1}_{\text{MIM}} \big], \qquad (1)$$

where $\phi_{txt} = g_{txt}^{de}(g_{txt}(f_{im}(I), f_{txt}(T_m)))$ and $\phi_{im} = g_{im}^{de}(g_{im}(f_{im}(I_m), f_{txt}(T)))$. The loss is computed only for masked pixels and text tokens. Hence, the superscript $M$ denotes data or features correspond to the masked signals. A pair of $I$ and $T$ is sampled from the training dataset $D$, $\mathcal{H}$ denotes cross-entropy, and $y_T^M$ is a matrix that contains one-hot row vectors for the ground truth of masked text tokens. $\Omega(\cdot)$ is the number of pixels. When minimizing $\mathcal{L}_{MVLM}$, the model is enforced to reconstruct the original signals by attending to the other modality signals. Cross-attending for reconstruction enables the model to learn interaction between V+L modalities.

## 3.2 Multi-modal Alignment

In addition to the masked signal modeling tasks, we adopt two additional tasks to explicitly learn multi-modality alignment. The first one is an image-text contrastive (ITC) learning (Radford et al., 2021; Jia et al., 2021). For the $k$-th pair of image and text features out of the image and text encoders, two separate FC layers are used to project the image [CLS] token features and the text [START] token features to the same dimensional feature space with unit norm, $z_{im}^k$ and $z_{txt}^k$, respectively. The loss, $\mathcal{L}_{ITC}$ is computed as

$$\mathcal{L}_{ITC} = -\frac{1}{N} \sum_{k=1}^N \left[ \log \frac{\exp(z_{im}^k \cdot z_{txt}^k / \tau)}{\sum_{n=1}^N \exp(z_{im}^k \cdot z_{txt}^n / \tau)} + \log \frac{\exp(z_{im}^k \cdot z_{txt}^k / \tau)}{\sum_{n=1}^N \exp(z_{im}^n \cdot z_{txt}^k / \tau)} \right], \qquad (2)$$

where $N$ and $\tau$ are the batch size and the temperature scaling parameter, respectively. The second task is an image-text matching (ITM) (Chen et al., 2020b; Li et al., 2021; 2020b), predicting whether an image and a text are aligned or not. The `[CLS]` and `[START]` token features from the image and text cross-modality encoders are $z_{im}^{cross}$ and $z_{txt}^{cross}$, respectively. To fuse these two features, we compute the element-wise product of $z_{im}^{cross}$ and $z_{txt}^{cross}$ ($z_{im}^{cross} * z_{txt}^{cross}$), and a FC layer followed by softmax is applied to obtain the final prediction (Lu et al., 2019). For training, we use $y_{ITM} = 1$, when $z_{im}^{cross}$ and $z_{txt}^{cross}$ are a pair. Otherwise, $y_{ITM} = 0$. The loss, $\mathcal{L}_{ITM}$, is defined as

$$\mathcal{L}_{ITM} = \mathbb{E}_{(I,T) \sim D}[\mathcal{H}(y_{ITM}, g^{itm}(z_{im}^{cross} * z_{txt}^{cross}))]. \tag{3}$$

Following (Li et al., 2021), we sample in-batch hard negatives based on the distribution of the cosine similarity between $z_{im}$ and $z_{txt}$. The overall pre-training loss, $\mathcal{L}$, is defined as $\mathcal{L} = \mathcal{L}_{MVLM} + \mathcal{L}_{ITC} + \mathcal{L}_{ITM}$. We term our model trained with loss $\mathcal{L}$ as MaskVLM (**Mask**ed **V**ision and **L**anguage **M**odeling).

### 3.3 PROBABILISTIC INTERPRETATION

We differentiate MaskVLM from the existing V+L models using masked signal modeling from a perspective of likelihood estimation. The training objective of masked signal modeling on uni-modal signals, $X$, focuses on learning the data distribution $p(X)$ which is formulated by the law of total probability as $p(X) = \sum_{X_m \in \mathcal{M}_X} p(X_m) \cdot p(X|X_m)$, where $X_m$ is an instance of masked signal from the set of all possible masked signals, $\mathcal{M}_X$. MIM or MLM learns the data distribution by maximizing $\sum_{X_m \in \mathcal{M}_X} p(X|X_m)$ (Bengio et al., 2013).

In V+L representation learning, the ultimate goal is to learn the joint distribution of multi-modal signals, $p(I, T)$. However, the authors in (Sohn et al., 2014) pointed out that directly maximizing the likelihood for the joint distribution is challenging because of the heterogeneous multi-modal data distributions. Instead, they show minimizing variation of information defined as $-\mathbb{E}_{(I,T) \sim D}(\log p(I|T) + \log p(T|I))$ is sufficient to estimate the joint distribution. From a perspective of variation of information, the limitations in existing works can be better understood. Several existing works attempted to approximate the joint distribution using MLM with unmasked image (Duan et al., 2022; Li et al., 2021; 2019; Yang et al., 2022). In other words, $p(T|I, T_m)$ is maximized to learn the conditional distribution, $p(T|I)$, but $p(I|T)$ is not modeled. In other existing works (Chen et al., 2020b; Li et al., 2020a; Lu et al., 2020; Su et al., 2019; Tan & Bansal, 2019), where both modalities are masked, the visual masking is limited to mask the visual features extracted from a frozen object detector, $\psi(\cdot)$, instead of the raw image pixels. In this case, the distributions $p(\psi(I)|T)$ and $p(T|\psi(I))$ are modeled instead of $p(I|T)$ and $p(T|I)$. This frozen feature extractor can bottleneck the direct estimation of the underlying data distribution. MaskVLM is trained end-to-end to estimate both conditional distributions, $p(I|T)$ and $p(T|I)$, which directly minimizes the variation of information. We hypothesize this modeling of conditional distributions for both modalities could lead to superior performance in both large-scale and limited data training scenarios, which we empirically demonstrated in Section 4.

## 4 EXPERIMENTS

### 4.1 PRE-TRAINING DATASETS AND DOWNSTREAM TASKS

We use the union of four datasets for pre-training so that we can perform a fair comparison with existing state-of-the-art methods (Chen et al., 2020b; Li et al., 2021). These datasets are Conceptual Captions (CC) (Sharma et al., 2018), SBU Captions (Ordonez et al., 2011), Visual Genome (VG) (Krishna et al., 2017), and COCO Captions (Lin et al., 2014). While VG and COCO contain captions annotated by humans, CC and SBU Captions are automatically collected from the web. The total number of unique images and image-text pairs in the four datasets are 4.1M and 5.2M, respectively. We term this pre-training dataset as the 4M dataset. We validate the pre-trained model on the following four downstream tasks:

**Image-Text Retrieval:** We perform text-to-image and image-to-text retrieval. We use the ITC and ITM losses of Section 3.2 for finetuning and the finetuned models are evaluated on COCO (Lin et al., 2014) and Flickr30k (Plummer et al., 2015). In addition, since COCO is used for pre-training, zero-shot retrieval performance is reported on Flickr30k. In (Li et al., 2021), the model finetuned on COCO is used for the zero-shot evaluation on Flickr30k. Although it may result in better performance, we believe that using the finetuned model does not validate the zero-shot capability of the

| Method | # images | MSCOCO (5K) | | | | | | Flickr30k (1K) | | | | | |
| | | Image Retrieval | | | Text Retrieval | | | Image Retrieval | | | Text Retrieval | | |
| | | R@1 | R@5 | R@10 | R@1 | R@5 | R@10 | R@1 | R@5 | R@10 | R@1 | R@5 | R@10 |
| ImageBERT (Qi et al., 2020) | 6M | 50.5 | 78.7 | 87.1 | 66.4 | 89.8 | 94.4 | 73.1 | 92.6 | 96.0 | 87.0 | 97.6 | 99.2 |
| UNITER (Chen et al., 2020b) | 4M | 52.9 | 79.9 | 88.0 | 65.7 | 88.6 | 93.8 | 75.6 | 94.1 | 96.8 | 87.3 | 98.0 | 99.8 |
| VILLA (Gan et al., 2020) | 4M | - | - | - | - | - | - | 76.3 | 94.2 | 96.8 | 87.9 | 97.5 | 98.8 |
| OSCAR (Li et al., 2020b) | 4M | 54.0 | 80.8 | 88.5 | 70.0 | 91.1 | 95.5 | - | - | - | - | - | - |
| ALBEF (Li et al., 2021) | 4M | 56.8 | 81.5 | 89.2 | 73.1 | 91.4 | 96.0 | 82.8 | **96.7** | **98.4** | 94.3 | **99.4** | 99.8 |
| Triple (Yang et al., 2022) | 4M | 59.0 | 83.2 | 89.9 | 75.6 | 92.8 | 96.7 | 84.0 | **96.7** | 98.5 | 94.9 | **99.5** | 99.8 |
| Codebook (Duan et al., 2022) | 4M | 58.7 | 82.8 | 89.7 | 75.3 | 92.6 | 96.6 | 83.3 | 96.1 | 97.8 | 95.1 | **99.4** | **99.9** |
| ALIGN (Jia et al., 2021) | 1.8B | 59.9 | 83.3 | 89.8 | 77.0 | 93.5 | 96.9 | 84.9 | 97.4 | 98.6 | 95.3 | 99.8 | 100.0 |
| MaskVLM | 4M | **60.1** | **83.6** | **90.4** | 76.3 | **93.8** | 96.8 | 84.5 | 96.7 | 98.2 | 95.6 | 99.4 | 99.9 |

Table 1: Comparison with finetuned state-of-the-art methods on image-text retrieval. The gray row indicates that the model is trained with significantly larger number of data than MaskVLM.

pre-trained model. Therefore, we use the pre-trained model directly for zero-shot evaluation. Following (Li et al., 2021), we first retrieve top-$k$ candidates using the similarity scores from the image and the text encoders. The top-$k$ candidates are further processed by the cross-modality encoders to obtain the final retrieval results.

**Visual Question Answering (VQA):** Here, given an image and a question pair, the model should generate a correct answer. The model is evaluated on VQA v2 (Goyal et al., 2017). We adopt the answer generation framework (Cho et al., 2021) and finetune the base model with a fusion encoder and an answer decoder. The model architectures are visualized in Figure 6 (a) of Appendix. The fusion encoder consists of one cross-attention block shown in Figure 3 (b). The output from the text cross-modality encoder is used as queries, and the image cross-modality encoder output is utilized to create attentions in the fusion encoder. The architecture of the answer decoder is the same as that of the text cross-modality encoder, but it is trained with a language modeling loss to generate the answers. Specifically, the output of the fusion encoder is used for computing attentions and the answer tokens are autoregressively predicted. During inference, [START] token is used as an initial token to generate following answer tokens.

**Natural Language for Visual Reasoning (NLVR):** This tasks involves a binary classification with a triplet, (text, image1, image2). The goal here is to predict whether the text describes the pair of images. For finetuning, we feedforward (text, image1) and (text, image2) separately to extract the features as shown in Figure 6 (b). The [CLS] token features of image1 and image2 from the image encoder are denoted as $v_1$ and $v_2$, respectively. The [START] token text features from the text encoder is $w$. These features are processed by the cross-modality encoders. The outputs of the image and text cross-modality encoders are fused by element-wise multiplication. The fused features for both images are concatenated, and a classifier with two linear layers predicts whether the text is aligned with the image pair or not. NLVR2 (Suhr et al., 2018) is used for the evaluation.

**Visual Entailment (VE):** Given an image text pair, the task is to classify the relationship between the image and the text into one of three categories: entailment, neutral, and contradictory. The element-wise product of the output from the image and the text cross-modality encoders is forwarded to a classifier of two linear layers for prediction. SNLI-VE (Xie et al., 2019) is used for evaluation.

## 4.2 IMPLEMENTATION DETAILS

We use a Visual Transformer (ViT) (Dosovitskiy et al., 2020) pre-trained on ImageNet (Deng et al., 2009) and a pre-trained RoBERTa from (Liu et al., 2019) to initialize the image and the text encoder, respectively. We pre-train the model for 50 epochs when the 4M dataset is used and 30 epochs for all other experiments. A batch size of 512 is used with 16 NVIDIA Tesla V100 GPUs. All parameters are optimized using AdamW (Loshchilov & Hutter, 2017) with a weight decay of 0.05. Following (Xie et al., 2021), we use the image masking ratio of 60%. While 15% masking ratio is used for text in language models (Devlin et al., 2018; Liu et al., 2019), we use 30% since the paired image can provide additional information for text reconstruction. During pre-training, the learning rate is warmed up to $3 \times 10^{-4}$ in the first 5 epochs and decayed to $3 \times 10^{-5}$ using a cosine scheduler. The learning rates for the image encoder and the text encoder are set to $10^{-5}$, which is less than that of the cross-modality encoders. An image size of $224 \times 224$ and RandAugment (Cubuk et al.,

| Method | # images | Flickr30k (1K) | | | | | |
| | | Image Retrieval | | | Text Retrieval | | |
| | | R@1 | R@5 | R@10 | R@1 | R@5 | R@10 |
|---|---|---|---|---|---|---|---|
| ImageBERT (Qi et al., 2020) | 6M | 54.3 | 79.6 | 87.5 | 70.7 | 90.2 | 94.0 |
| Unicoder-VL (Li et al., 2020a) | 3.8M | 48.4 | 76.0 | 85.2 | 64.3 | 85.8 | 92.3 |
| ViLT (Kim et al., 2021) | 4M | 55.0 | 82.5 | 89.8 | 73.2 | 93.6 | 96.5 |
| UNITER (Chen et al., 2020b) | 4M | 66.2 | 88.4 | 92.9 | 80.7 | 95.7 | 98.0 |
| ALBEF (Li et al., 2021) | 4M | 68.2 | 88.6 | 93.0 | 84.9 | 97.2 | 99.0 |
| FLAVA (Singh et al., 2022) | 68M | 65.2 | 89.4 | - | 67.7 | 94.0 | - |
| CLIP (Radford et al., 2021) | 400M | 68.7 | 90.6 | 95.2 | 88.0 | 98.7 | 99.4 |
| ALIGN (Jia et al., 2021) | 1.8B | 75.7 | 93.8 | 96.8 | 88.6 | 98.7 | 99.7 |
| MaskVLM | 4M | **75.0** | **92.5** | **95.8** | **87.0** | **97.9** | **99.3** |

Table 2: Zero-shot image-text retrieval performance on Flickr30k. The gray row indicates that the model is trained with significantly larger number of data than MaskVLM.

| Method | # images | VQA | | NLVR2 | | SNLI-VE | |
| | | test-dev | test-std | dev | test-P | val | test |
|---|---|---|---|---|---|---|---|
| VisualBERT (Li et al., 2019) | 113K | 70.80 | 71.00 | 67.40 | 67.00 | - | - |
| VL-BERT (Su et al., 2019) | 3.3M | 71.16 | - | - | - | - | - |
| LXMERT (Tan & Bansal, 2019) | 180K | 72.42 | 72.54 | 74.90 | 74.50 | - | - |
| 12-in-1 (Lu et al., 2020) | 3.3M | 73.15 | - | - | 78.87 | - | 76.95 |
| UNITER (Chen et al., 2020b) | 4M | 72.70 | 72.91 | 77.18 | 77.85 | 78.59 | 78.28 |
| VL-BART/T5 (Cho et al., 2021) | 180K | - | 71.30 | - | 73.60 | - | - |
| ViLT (Kim et al., 2021) | 4M | 70.94 | - | 75.24 | 76.21 | - | - |
| OSCAR (Li et al., 2020b) | 4M | 73.16 | 73.44 | 78.07 | 78.36 | - | - |
| VILLA (Gan et al., 2020) | 4M | 73.59 | 73.67 | 78.39 | 79.30 | 79.47 | 79.03 |
| ALBEF (Li et al., 2021) | 4M | 74.54 | 74.70 | 80.24 | 80.50 | 80.14 | 80.30 |
| Triple (Yang et al., 2022) | 4M | 74.90 | 74.92 | 80.54 | 80.11 | **80.51** | 80.29 |
| Codebook (Duan et al., 2022) | 4M | 74.86 | 74.97 | 80.50 | 80.84 | **80.47** | 80.40 |
| FLAVA (Singh et al., 2022) | 68M | 72.80 | - | - | - | - | 79.00 |
| SimVLM$_{base}$ (Wang et al., 2021) | 1.8B | 77.87 | 78.14 | 81.72 | 81.77 | 84.20 | 84.15 |
| MaskVLM | 4M | **75.45** | **75.40** | **81.58** | **81.98** | **80.37** | **80.67** |

Table 3: Comparison with state-of-the-art methods on VQA, NLVR2, and VE. The gray row indicates that the model is trained with significantly larger number of data than MaskVLM.

2020) are used. During finetuning, the image is resized to $384 \times 384$ and the positional encoding is interpolated following (Dosovitskiy et al., 2020). More details can be found in Appendix.

## 4.3 Evaluation on Image-Text Retrieval, VQA, NLVR, and VE

We compare the finetuned image-text retrieval performance of the proposed MaskVLM with the state-of-the-art methods in Table 1. The second column is the number of unique images used for pretraining and the retrieval performance is evaluated in terms of Recall@k (R@k). We do not directly compare with ALIGN (Jia et al., 2021) since it is trained with more than 300 times of data used for MaskVLM. However, we still highlight the small performance gap between MaskVLM and ALIGN. We achieve the best performance in all Recall@k metrics on both COCO and Flickr30k except for the image retrieval R@10 and text retrieval R@5 on Flickr30k. Compared to ALIGN, we even achieve higher R@1 for image retrieval on COCO and text retrieval on Flickr30k. Table 2 shows the zero-shot retrieval performance of the state-of-the-art methods on Flickr30k. MaskVLM achieves a significant improvement over the second best method, ALBEF (Li et al., 2021), by 6.8 points at R@1 for image retrieval. Given that ALBEF is not trained for MIM, we hypothesize that ALBEF achieves the biased performance for text retrieval and MaskVLM achieves the significant improvement in image retrieval by additional MIM which models $p(I|T)$. While FLAVA exploits both MLM and MIM with the pre-trained image tokenizer, using 13 times more data than MaskVLM, MaskVLM still outperforms FLAVA by 9.8 and 19.3 points at R@1 for image and text retrieval respectively. Compared with CLIP (Radford et al., 2021) which is trained with at least 76 times more data than MaskVLM, we still achieve higher R@1 for image retrieval by 6.3 points. In general, MaskVLM achieves state-of-the-art performance in both finetuning and zero-shot experiments.

We report the accuracies on VQA, NLVR, and VE in Table 3. Except for SimVLM whose pretraining data is more than 300 times larger than that of MaskVLM, we consistently achieve the best

| Dataset (# of samples) | Method | COCO IR | | COCO TR | | VQA | NLVR2 | | SNLI-VE | |
|---|---|---|---|---|---|---|---|---|---|---|
| | | R@1 | R@5 | R@1 | R@5 | dev | dev | test-P | val | test |
| CC 50% + COCO (2M) | ALBEF | 49.97 | 77.35 | 65.76 | 89.32 | 73.07 | 76.58 | 76.89 | 79.20 | 79.19 |
| | Ours | **56.36** | **81.98** | **73.22** | **92.00** | **74.24** | **79.81** | **79.47** | **79.69** | **79.50** |
| CC 25% + COCO (1.3M) | ALBEF | 48.09 | 75.63 | 64.24 | 87.20 | 72.65 | 75.20 | 76.78 | 78.84 | 78.96 |
| | Ours | **55.11** | **81.28** | **72.18** | **91.48** | **74.17** | **79.13** | **78.94** | **79.35** | **79.76** |
| CC 10% + COCO (0.9M) | ALBEF | 45.33 | 73.57 | 61.00 | 84.98 | 72.14 | 74.81 | 74.64 | 78.51 | 78.36 |
| | Ours | **54.04** | **80.74** | **70.04** | **91.24** | **73.93** | **78.62** | **77.19** | **79.11** | **79.60** |

Table 4: Downstream task performance with limited pre-training data.

performances in all these tasks except for the validation split of NLVR2. In particular, MaskVLM is better than the second best method by 0.43, 1.14, and 0.27 on the test splits of VQA, NLVR2, and SNLI-VE, respectively. Compared to the base model of SimVLM, we narrow the accuracy gaps to 2.74% and 3.48% in test-std and test splits of VQA and VE, respectively. MaskVLM achieves higher accuracy than SimVLM$_{base}$ in the test split of NLVR2 by 0.21%.

## 4.4 EVALUATION WITH LIMITED PRE-TRAINING DATA

We highlight the performance of MaskVLM in limited data scenarios. In particular, we create three subsets of the 4M pre-training data by sampling 50%, 25%, and 10% of CC and combining them with COCO. The number of image-text pairs in each subset is around 39%, 25%, and 16% of the 4M pre-training data which contain 5.2M pairs, respectively. We pre-train models with these subsets of the data and analyze the downstream task performance in comparison with state-of-the-art methods. The results are reported in Table 4. Particularly, image and text retrieval R@1 performance on COCO is also visualized in Figure 4. We compare MaskVLM with

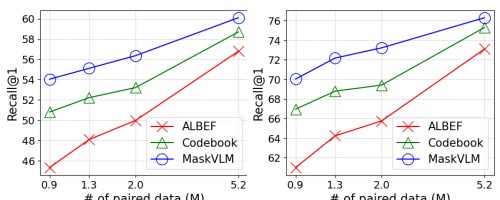

Figure 4: R@1 plots for image retrieval (left) and text retrieval (right) on COCO using limited pre-training data.

the most recent state-of-the-art methods, which are ALBEF (Li et al., 2021) and Codebook (Duan et al., 2022). In Table 4, as the size of pre-training data becomes smaller from CC 50% + COCO to CC 10% + COCO, the performance gap between MaskVLM and ALBEF increases from 6.39 to 8.71 at R@1 in COCO image retrieval (IR), 7.46 to 9.04 at R@1 in COCO text retrieval (TR), 1.17 to 1.79 in VQA and 0.31 to 1.24 in the test set of SNLI-VE. In NLVR2 and VQA, MaskVLM trained with CC 10% + COCO achieves higher accuracy than ALBEF trained with CC50% + COCO, which contains more than twice of image-text pairs in CC 10% + COCO. In Figure 4, while Codebook shows competitive recall performance compared to the MaskVLM with the 4M dataset ( 5.2M pairs), the R@1 differences in image and text retrieval, respectively, increase from 1.4 and 1.0 in the 4M dataset to 3.15 and 3.80 in CC 50% + COCO. Our model trained with CC25% + COCO outperforms Codebook trained with CC50% + COCO by 1.90 and 2.76 points in terms of image and text retrieval R@1, respectively. Since one of the main differences in MaskVLM compared to ALBEF and Codebook is the additional MIM, we believe that joint modeling of V+L contribute to better performance in limited data scenarios.

## 4.5 ABLATION STUDY

We perform an ablation study using different combinations of loss components to highlight the contribution of masked V+L modeling. We compare six models with the same architecture but with different loss functions for pre-training. We pre-train all models on the CC 50% + COCO dataset and compare finetuned and zero-shot retrieval performance on Flickr30k in Table 5. We note that zero-shot evaluation of the MLM + MIM model cannot be performed because the FC layers to compute ITM and ITC are not trained during pre-training. ITC and ITM are closely related to the retrieval task since they are used for finetuning and measuring the similarity between images and texts. However, MLM + MIM still achieves significantly better finetuned and zero-shot performance than ITC, which shows that MLM + MIM alone learns meaningful V+L representations. Compared to ITC+ ITM in the finetuned performance, ITC + ITM + MLM achieves slightly improved R@1

| Loss | Finetuned | | | | Zero-shot | | | |
|------|-----------|--|--|--|-----------|--|--|--|
| | IR | | TR | | IR | | TR | |
| | R@1 | R@5 | R@1 | R@5 | R@1 | R@5 | R@1 | R@5 |
| ITC | 65.10 | 89.88 | 80.10 | 96.90 | 55.08 | 80.90 | 68.40 | 90.00 |
| ITC + ITM | 79.96 | 95.56 | 92.30 | 98.90 | 69.50 | 89.54 | 82.40 | 96.60 |
| MLM +MIM | 76.08 | 94.40 | 90.30 | 98.80 | - | - | - | - |
| ITC + ITM + MLM | 80.34 | 95.82 | 92.00 | 99.30 | 70.74 | 90.92 | 84.40 | 97.10 |
| ITC + ITM + MIM | 80.12 | 95.56 | 91.50 | 99.00 | 69.26 | 90.30 | 82.90 | 97.20 |
| ITC + ITM + MLM + MIM | **81.26** | **96.00** | **94.10** | **99.60** | **71.18** | **91.12** | **85.60** | **97.50** |

Table 5: Image-text retrieval evaluation on Flickr30k with different loss functions for pre-training.

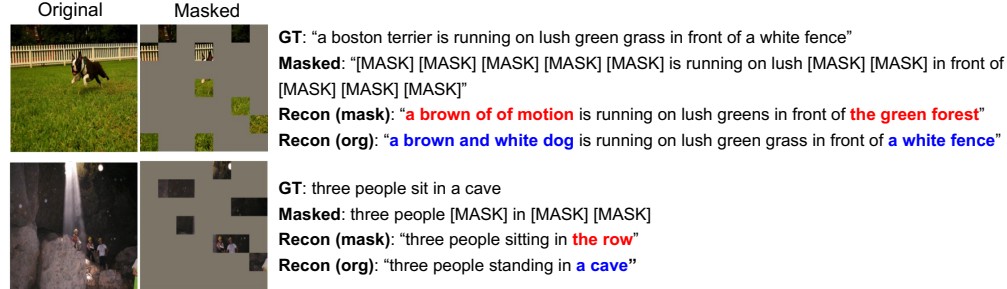

Figure 5: Masked language modeling examples using masked and original images. "Recon (mask)" and "Recon (org)" denote reconstructed text from the masked image and the original image, respectively.

by 0.38 in image retrieval and degraded R@1 by 0.3 in text retrieval. When MIM alone is used with ITC + ITM as well, finetuned R@1 is improved by 0.16 and degraded by 0.8 for image and text retrieval, respectively, over ITC + ITM. On the other hand, when ITC + ITM + MLM + MIM is used, the model achieves significant improvement of finetuned performance over ITC + ITM + MLM by 0.92 and 2.10 for R@1 image and text retrieval, respectively. ITC + ITM + MLM + MIM also obtains the best performance in zero-shot retrieval. This result further supports the advantage of joint modeling for masked V+L signals.

## 4.6 Qualitative Results

We perform a qualitative analysis to show the role of multi-modal information in the reconstruction of masked signals from our model. To be specific, we illustrate the prediction of masked text tokens with and without the corresponding images. This illustration highlights how MaskVLM effectively utilizes both modality information to complete the masked signal modeling task. Figure 5 shows the reconstruction of masked texts using original images ("Recon (org)") and masked images ("Recon (mask)"). In the first top example, when the model is given a masked text and a masked image which does not contain the "dog", the reconstruction is performed by using only available information such as image patches of "green grass". Thus, the prediction is limited to "a brown of motion" or "the green forest". However, when the original image is used for reconstruction, both "a brown and white dog" and "white fence" are reconstructed by accurately attending to the image. In the bottom example, the visible patches of the masked image contain a few people, but lack background information. Consequently, the reconstruction with the masked image does not contain any background information but the background "cave" is reflected in the reconstruction with the original image. Theses examples confirm that MaskVLM has learned to perform masked signal modeling using both V+L information.

## 5 Conclusion

We propose masked vision and language modeling as a pre-training task for learning V+L representations. We provide a probabilistic interpretation to highlight the contribution of the proposed method and validate its advantages in both large-scale and limited data regimes. We consistently achieve the state-of-the-art performance in a broad range of V+L tasks.

ACKNOWLEDGMENTS

We thank Jiali Duan for providing results of the Codebook (Duan et al., 2022) with limited pre-training data in Figure 4.

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

# A APPENDIX

## A.1 DETAILS ON FINETUNING FOR DOWNSTREAM TASKS

We explain implementation details for each of the downstream tasks. For all the downstream tasks, we use AdamW (Loshchilov & Hutter, 2017) with a weight decay of 0.05 and the cosine scheduler. An image size of $384 \times 384$ with RandAugment (Cubuk et al., 2020) is utilized and the positional encoding is interpolated following (Dosovitskiy et al., 2020). Except for the VQA task, we use the model achieves the best performance in the validation set to report the performance on the test set. We use the last epoch model for the VQA evaluation.

**Image-Text Retrieval:** COCO (Lin et al., 2014) and Flickr30k (Plummer et al., 2015) are used to report the performance. To be specific, we follow data splits proposed in (Karpathy & Fei-Fei, 2015) and an average recall over image and text retrieval is used to find the best model in the validation set. The pre-trained model is finetuned for 15 epochs with a batch size of 256 and a learning rate of $1 \times 10^{-5}$.

**Visual Question Answering (VQA):** For a fair comparison with existing methods (Chen et al., 2020b; Li et al., 2021), we use training and validation sets from VQA v2.0 (Goyal et al., 2017) with a subset of VQA samples from Visual Genome (Krishna et al., 2017) for training. Also, we report performance on both test-dev and test-std splits of VQA v2.0. Following (Li et al., 2021), we weight the loss for each answer based on its occurrence among all the answers. The model is finetuned for 15 epochs with a batch size of 256 . We use a learning rate of $2 \times 10^{-5}$ for the image and the text cross-modality encoders, the fusion encoder, and the answer decoder. For the image and the text encoders, a learning rate of $1 \times 10^{-5}$ is used. The fusion encoder and the answer decoder are initialized by the last and all three blocks of the pre-trained text cross-modality encoder, respectively.

**Natural Language for Visual Reasoning (NLVR):** Data splits proposed in (Suhr et al., 2018) are used for finetuning and evaluation. The model is finetuned for 5 epochs with a batch size of 128. Since the classifier is newly added after finetuning, we use a learning rate of $1 \times 10^{-4}$ for the classifier and $1 \times 10^{-5}$ for the remaining parts of the model. Different from (Duan et al., 2022; Li et al., 2021; Yang et al., 2022), where the models require additional text-assignment pre-training step before finetuning, we directly finetune for simplicity.

**Visual Entailment (VE):** We follow data splits proposed in SNLI-VE (Xie et al., 2019). We finetune the model with a batch size of 256 for 5 epochs. Similar to the NLVR task, a learning rate of $1 \times 10^{-4}$ is used for the classifier and $1 \times 10^{-5}$ is used for the remaining parts of the model.

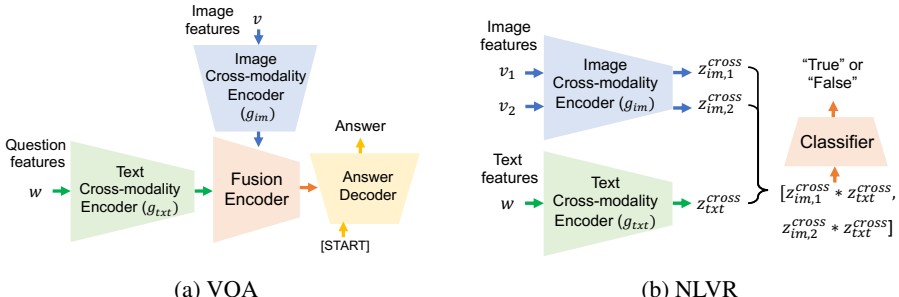

Figure 6: An illustration of model architectures for VQA and NLVR.

## A.2 REPRODUCIBILITY

We add more details of MaskVLM for reproducibility. We used the ImageNet pretrained ViT (`vit_base_patch16_224`) from (Wightman, 2019) and the pre-trained RoBERTa (`roberta-base`) from Hugging Face (Wolf et al., 2020). The detailed model architectures are visualized in Figure 7. Following (Dosovitskiy et al., 2020), the image encoder uses layer normalization (Ba et al., 2016) before each multi-head attention block while the text encoder applies layer normalization after each multi-head attention block (post norm). For the image (text) cross-modality encoder, we adopt the post norm and use the outputs of the text (image) encoder as keys and values

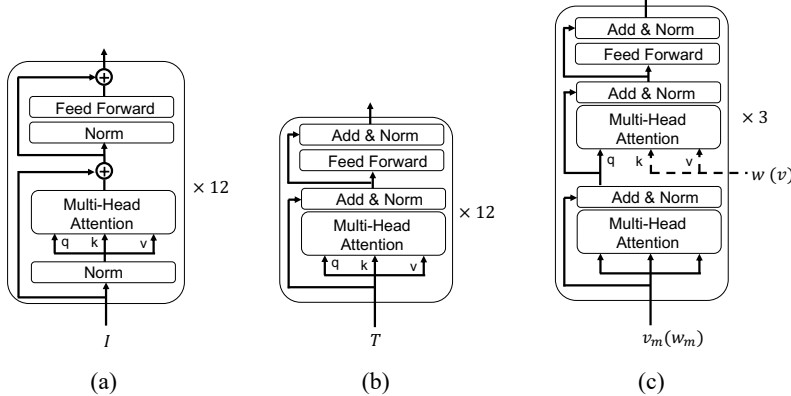

Figure 7: Model architectures of (a) Image encoder, (b) Text encoder and (c) Image (text) cross-modality encoder. The dotted lines in (c) denote key and value from the other modality for cross-attention.

| Method | MSCOCO (5K) | | | | | | Flickr30k (1K) | | | | | |
|---|---|---|---|---|---|---|---|---|---|---|---|---|
| | Image Retrieval | | | Text Retrieval | | | Image Retrieval | | | Text Retrieval | | |
| | R@1 | R@5 | R@10 | R@1 | R@5 | R@10 | R@1 | R@5 | R@10 | R@1 | R@5 | R@10 |
| ALBEF | 56.8 | 81.5 | 89.2 | 73.1 | 91.4 | 96.0 | 82.8 | 96.7 | 98.4 | 94.3 | 99.4 | 99.8 |
| MaskVLM (both) | **59.5** | 83.4 | 90.2 | **76.0** | 93.4 | 96.9 | **83.9** | 96.6 | 98.3 | **95.1** | 99.9 | 100.0 |
| MaskVLM (one) | **60.1** | 83.6 | 90.4 | **76.3** | 93.8 | 96.8 | **84.5** | 96.7 | 98.2 | **95.6** | 99.4 | 99.9 |

Table 6: Comparison of finetuned MaskVLMs with different masking strategies and ALBEF on image-text retrieval. (MaskVLM (One): masking one modality at a time for computing MLM and MIM losses. MaskVLM (both): masking both modalities at the same time for reconstruction)

to compute cross-attention. To compute MIM and MLM, the self-attention outputs of the masked image features, $v_m$, is used as queries and the unmasked text features, $w$, are used as keys and values in the image cross-modality encoder. For the text cross-modality encoder, the masked text features, $w_m$, are used as queries and the unmasked image features, $v$, are used as keys and values. To keep the framework simple, we do not use any loss weighting for each loss term and layer decay during finetuning.

### A.3   ABLATION STUDY ON MASKING STRATEGIES

We study different masking strategies in computing MIM and MLM losses. In particular, we compare MaskVLM using one modality masked and the other modality unmasked for reconstruction (MaskVLM (one)) and MaskVLM using both modalities masked at the same time for reconstruction. We compare these two MaskVLM models with the state-of-the-art method, ALBEF in Table 6. We follow the experimental setup described in Section 4.1 and report the finetuning performance on image-text retrieval. The performance of masking one modality at a time (MaskVLM (one)) was slightly better than masking both modalities at the same time (MaskVLM (both)). However, we observed that both reconstruction strategies are still effective as they achieve higher R@1 for image and text retrieval in both COCO and Flickr30k compared to ALBEF.

### A.4   ABLATION STUDY ON MASKING RATIO

We perform ablation study using different masking ratios for masked vision and language modeling. In particular, we pre-train MaskVLM with several combinations of image and text masking ratios on the CC 50% + COCO dataset and report the finetuned image-text retrieval performance on Flickr30k in Table 7. We also report an average of R@k for image and text retrieval. When only image masking ratio is changed in the first three rows of the table, the difference between the maximum and the minimum of the average recall is 0.26 for image retrieval and 0.10 for text retrieval. This shows that MaskVLM achieves stable performance across the tested image masking ratios. From

comparison between the second row and the last row, we observe that increasing the text masking ratio from 0.15 to 0.3 leads to higher recall performance for both image and text retrieval.

| Masking ratio (image / text) | Image Retrieval | | | | Text Retrieval | | | |
|---|---|---|---|---|---|---|---|---|
| | R@1 | R@5 | R@10 | Average | R@1 | R@5 | R@10 | Average |
| 0.5 / 0.3 | 81.32 | 96.04 | 97.92 | 91.76 | 93.30 | 99.70 | 100.00 | 97.67 |
| 0.6 / 0.3 | 81.26 | 96.00 | 97.78 | 91.68 | 94.10 | 99.60 | 99.60 | 97.77 |
| 0.7 / 0.3 | 81.82 | 96.00 | 98.00 | 91.94 | 93.60 | 99.50 | 99.90 | 97.67 |
| 0.6 / 0.15 | 80.30 | 95.66 | 97.82 | 91.26 | 92.50 | 99.10 | 99.60 | 97.07 |

Table 7: Finetuned image-text retrieval performance on Flickr30k with different masking ratios for masked vision and language modeling.

## A.5 EVALUATION ON IMAGE RECOGNITION

We evaluate the image recognition performance of MaskVLM. Following CLIP (Radford et al., 2021), we perform image classification directly using the pre-trained MaskVLM on various image recognition datasets including UC Merced Land Use (Yang & Newsam, 2010), MIT-67 (Quattoni & Torralba, 2009), CUB-200 (Wah et al., 2011), Oxford Flowers (Nilsback & Zisserman, 2008), Caltech-256 (Griffin et al., 2007), and ImageNet-1K (Deng et al., 2009). We compare the Top-1 accuracy of MaskVLM with ALBEF (Li et al., 2021) in Table 8. Both models are pre-trained with the 4M dataset. Since during pre-training stage, both MaskVLM and ALBEF were initialized with the ImageNet pre-trained weights, the evaluation on ImageNet is not strictly zero-shot but the evaluation on other datasets is. We formulate image classification as image-to-text retrieval where the similarity scores between a query image and all the class names are computed to retrieve top-1 class name. The similarity scores can be obtained using either ITC and ITM scores, and we report them separately. Also, the results of using prompt engineering as in CLIP are reported.

As shown in Table 8, MaskVLM consistently outperforms ALBEF across all the datasets. In particular, prompt engineering improves the average accuracy across all the datasets for MaskVLM but ALBEF achieves lower average accuracy with prompt engineering. This shows that MaskVLM can better align images with variants of text than ALBEF, which results in stronger V+L representations of MaskVLM for the image recognition task.

| Method | Prompt Engineering | UC Merced Land Use | MIT-67 | CUB-200 | Oxford Flowers | Caltech256 | ImageNet-1K | Average |
|---|---|---|---|---|---|---|---|---|
| ALBEF (ITC) | ✗ | 31.62 | 52.46 | 5.78 | 26.93 | 45.75 | 36.28 | 33.14 |
| ALBEF (ITM) | ✗ | 26.29 | 56.19 | 5.68 | 21.68 | 41.79 | 35.41 | 31.17 |
| ALBEF (ITC) | ✓ | 28.38 | 51.04 | 5.33 | 25.44 | 48.71 | 30.26 | 31.53 |
| ALBEF (ITM) | ✓ | 15.81 | 29.70 | 3.90 | 18.86 | 26.57 | 12.53 | 17.90 |
| MaskVLM (ITC) | ✗ | 41.52 | 55.60 | 4.97 | 26.61 | 60.68 | 33.16 | 37.09 |
| MaskVLM (ITM) | ✗ | 29.52 | 58.21 | **10.36** | 36.14 | 58.95 | 34.59 | 37.96 |
| MaskVLM (ITC) | ✓ | **45.14** | **66.27** | 4.37 | 32.72 | 60.95 | 39.16 | 41.43 |
| MaskVLM (ITM) | ✓ | 40.76 | 63.36 | 8.53 | **44.53** | **61.77** | **39.71** | **43.11** |

Table 8: Top-1 accuracy of pre-trained MaskVLM and ALBEF on image recognition. ITC and ITM denote the alignment scores utilized to perform image classification.

## A.6 ADDITIONAL EXAMPLES FOR THE QUALITATIVE ANALYSIS

We present additional examples for the qualitative analysis of MaskVLM in Figure 8. Similar to Figure 5, masked text tokens are reconstructed with masked images ("Recon (mask)") and original images ("Recon (org)"). We highlight that MaskVLM utilizes both V+L information to reconstruct the text which corresponds to the given image.

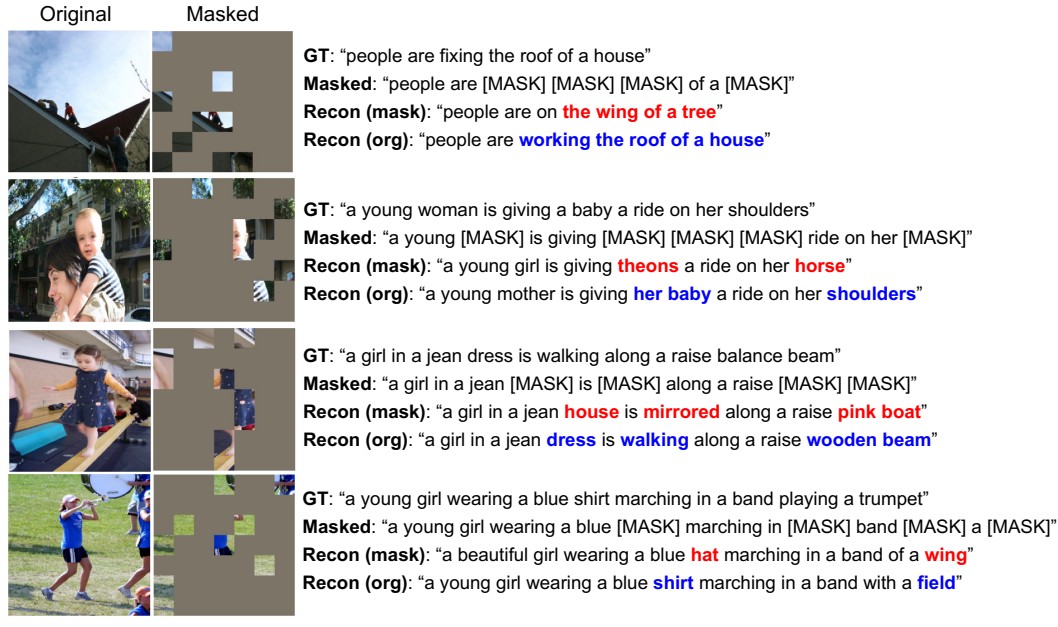

Original    Masked

**GT**: "people are fixing the roof of a house"
**Masked**: "people are [MASK] [MASK] [MASK] of a [MASK]"
**Recon (mask)**: "people are on **the wing of a tree**"
**Recon (org)**: "people are **working the roof of a house**"

**GT**: "a young woman is giving a baby a ride on her shoulders"
**Masked**: "a young [MASK] is giving [MASK] [MASK] [MASK] ride on her [MASK]"
**Recon (mask)**: "a young girl is giving **theons** a ride on her **horse**"
**Recon (org)**: "a young mother is giving **her baby** a ride on her **shoulders**"

**GT**: "a girl in a jean dress is walking along a raise balance beam"
**Masked**: "a girl in a jean [MASK] is [MASK] along a raise [MASK] [MASK]"
**Recon (mask)**: "a girl in a jean **house** is **mirrored** along a raise **pink boat**"
**Recon (org)**: "a girl in a jean **dress** is **walking** along a raise **wooden beam**"

**GT**: "a young girl wearing a blue shirt marching in a band playing a trumpet"
**Masked**: "a young girl wearing a blue [MASK] marching in [MASK] band [MASK] a [MASK]"
**Recon (mask)**: "a beautiful girl wearing a blue **hat** marching in a band of a **wing**"
**Recon (org)**: "a young girl wearing a blue **shirt** marching in a band with a **field**"

Figure 8: Additional masked language modeling examples using masked and original images. "Recon (mask)" and "Recon (org)" denote reconstructed text using the masked image and the original image, respectively.

## A.7 STATISTICS OF THE PRE-TRAINING DATASET

In Table 9, we report the statistics of the 4M pre-training dataset that MaskVLM is trained on. We note that some data urls provided in the web datasets can become invalid, which may lead to slightly different number of image-text pairs depending on when the datasets are downloaded.

| Dataset | # of image-text pairs | # of images |
|---------|----------------------|-------------|
| COCO    | 566,747              | 113,287     |
| CC      | 2,912,317            | 2,912,317   |
| SBU     | 1,000,000            | 1,000,000   |
| VG      | 768,536              | 100,406     |
| **Total** | **5,247,600**      | **4,126,010** |

Table 9: Statistics of the 4M pre-training dataset.

