# OpenReview forum: "Masked Vision and Language Modeling for Multi-modal Representation Learning"
_ICLR.cc/2023/Conference — ICLR 2023 poster_

### Official Review · Reviewer_cmqf · 2022-10-16

**Confidence:** 4
**Correctness:** 3
**Technical Novelty And Significance:** 2
**Empirical Novelty And Significance:** 2
**Recommendation:** 5

**Clarity, Quality, Novelty And Reproducibility:**

The paper is well-written and clear. The method is simple which I think is a strength, and this additionally makes the exposition clear.

The paper relies on public datasets which aid in reproducibility – I’d like to ask if the authors plan to open-source the code. One potential issue with reproducibility is the lack of error bars in the comparisons. Could error bars be added at least for the fine-tuning tasks? This would ensure that the improvements are not statistical outliers.

A significant issue with the paper is that several closely related references are missing. Here are a few:

* https://arxiv.org/abs/2208.10442
* https://arxiv.org/abs/2205.14204
* https://arxiv.org/abs/2204.01678

The novelty of the proposed method should be discussed in light of these previous papers which also focus on masked training for VL tasks. Some of these methods should also be added to the tables to ensure that the method is compared to the latest work.



**Strength And Weaknesses:**

Strengths:

* The proposed method is simple and intuitive.
* The paper relies on open-source datasets which significantly improves reproducibility.
* The paper shows strong improvements over baseline methods.

Weaknesses:

* There are multiple missing references. The novelty of the proposed method should be discussed in light of these.
* Given that there are missing references, the baselines in the tables might need to be revised. Thus, it is not clear how well the model performs compared to SOTA methods.
* Error bars are missing.


**Summary Of The Paper:**

This paper considers training vision-language models by masked reconstruction. Specifically, the model is trained to reconstruct a masked image conditioned on the paired text, and vice versa. In addition to this task, the standard contrastive and ITM tasks are added during pretraining. The authors use public small-scale VL datasets for pretraining and beat baseline methods on retrieval and VQA/NLVR-type tasks. The authors also show that the proposed method works well in low-data regimes and that all tasks contribute to the final performance.



**Summary Of The Review:**

The paper is well-written and presents a clear and intuitive idea that works well empirically. The authors use public datasets which improve reproducibility but there seem to be no error bars. The biggest issue is that multiple closely related papers which also focus on masked training for VL models are missing. When accounting for these papers, the novelty of the paper is decreased and the numbers in the tables might also need to be revised,

---

> ### Author Response · Authors · 2022-11-19
> **Reply to Reviewer cmqf (1)**
>
> We thank the reviewer for the constructive comments. Please find our answers to your questions below.
>
> \
> **Q1: Missing references of the related and concurrent works**
>
> A. We would like to note that according to ICLR 2023 review guidelines, our submission should not be penalized by not citing papers which are published after May 28, 2022 or not published in peer-reviewed venues (e.g. arXiv). **The mentioned related works, BEiT-3, M3AE, are only available on arXiv. Also, M3AE is the concurrent submission to ICLR 2023. MutilMAE was published at ECCV 2022 where the final acceptance decision was notified on July 3, 2022 which is after May 28, 2022.** Below, we quoted the review guideline from the official ICLR 2023 website for the reviewer’s convenience.
>
> >*Are authors expected to cite and compare with very recent work? What about non peer-reviewed (e.g., ArXiv) papers? (updated on 7 November 2022)*
> >
> >*“We consider papers contemporaneous if they are published (available in online proceedings) within the last four months. That means, since our full paper deadline is September 28, if a paper was published (i.e., at a peer-reviewed venue) on or after May 28, 2022, authors are not required to compare their own work to that paper. Authors are encouraged to cite and discuss all relevant papers, but they may be excused for not knowing about papers not published in peer-reviewed conference proceedings or journals, which includes papers exclusively available on arXiv....”*
>
> * Comparison with the concurrent works
>
> Although, per ICLR 2023 review policy, it is not mandatory to cite most of the mentioned papers, we would like to discuss them in the revision, and we believe MaskVLM has sufficient differences in comparison with the related recent and concurrent works (BEiT-3, M3AE, MultiMAE).
>
> **First**, MaskVLM is not dependent on any form of pre-trained image tokenizer but BEiT-3 is. BEiT-3 performs MIM by predicting image tokens for the masked patches. Since image tokens need to be obtained, BEiT-3 utilizes an image tokenizer trained by distilling knowledge from the CLIP model trained with 400 million image-text pairs. However, MaskVLM performs MIM by predicting masked pixels directly and does not require the image tokenizer trained with additional data.
>
> **Second**, MultiMAE is not trained with the language data but with different vision modality data (e.g. RGB, depth, semantic segmentation). Also, both MultiMAE and M3AE are not evaluated on V+L downstream tasks but only on vision downstream tasks. MaskVLM is trained to align V+L data and also evaluated on diverse V+L downstream tasks.
>
> BEiT-3 is trained with a combination of uni-modal data and 35M image-text pairs which is about 7 times larger than what MaskVLM used. BEiT-3 has total 1.9B trainable model parameters which is more than 50 times larger than MaskVLM (~336M model parameters). Considering these difference in pre-training data and model size, direct comparison of experimental results between BEiT-3 and MaskVLM is not fair. Also, since MultiMAE and M3AE did not report results on V+L tasks, they cannot be added to the tables in the Experiments section.
>
> Compared to BEiT-3, MultiMAE, and M3AE, we highlight the novelty of MaskVLM in reconstructing masked pixels directly without trained image tokenizers and evaluation on diverse V+L tasks in the Related Work section as follows:
>
> [Related Work]
>
> *“In (Bachmann et al., 2022), multiple vision modality data are masked and reconstructed to learn visual representations.”*
>
> *“While both MIM and MLM are explored in (Geng et al., 2022), the trained model is evaluated only on vision tasks. In (Dou et al., 2022; Fu et al., 2021; Singh et al., 2022; Wang et al., 2022), image tokens defined by image tokenizers trained with additional 250 million images (Ramesh et al., 2021) or distillation from the CLIP model (Radford et al., 2021) trained with 400 million image-text pairs (Peng et al., 2022) are reconstructed. In our work, we eliminate these model and data dependencies, by directly recovering RGB pixels and text tokens from masked image patches and masked text tokens. Therefore, MIM and MLM are seamlessly integrated to achieve generalizable V+L representations within a simple training framework.”*
>
> [References]
>
> *Roman Bachmann, David Mizrahi, Andrei Atanov, and Amir Zamir. Multimae: Multi-modal multi-task masked autoencoders. arXiv preprint arXiv:2204.01678, 2022.*
>
> *Xinyang Geng, Hao Liu, Lisa Lee, Dale Schuurams, Sergey Levine, and Pieter Abbeel. Multimodal masked autoencoders learn transferable representations. arXiv preprint arXiv:2205.14204, 2022.*
>
> *Wenhui Wang, Hangbo Bao, Li Dong, Johan Bjorck, Zhiliang Peng, Qiang Liu, Kriti Aggarwal, Owais Khan Mohammed, Saksham Singhal, Subhojit Som, et al. Image as a foreign language: Beit pretraining for all vision and vision-language tasks. arXiv preprint arXiv:2208.10442, 2022.*

---

> > ### Author Response · Authors · 2022-11-19
> > **Reply to Reviewer cmqf (2)**
> >
> > **Q2: The paper relies on public datasets which aid in reproducibility – I’d like to ask if the authors plan to open-source the code. One potential issue with reproducibility is the lack of error bars in the comparisons. Could error bars be added at least for the fine-tuning tasks? This would ensure that the improvements are not statistical outliers.**
> >
> > A: We plan to make the code publicly available upon acceptance of this paper. Following most of the state-of the-methods (e.g. UNITER, ALBEF, OSCAR, to mention a few), we did not report the error bars in the original submission, and used one fixed random seed for all experiments to ensure the reproducibility. We agree that it is a good idea to include the error bar for reproducibility purpose. However, our computing resources are allocated to the pre-training experiments of the 14M dataset requested by another reviewer and we could not provide error bars at this rebuttal stage. We will try our best to have some error bars ready at least for a few finetuning tasks during the second discussion phase.
> >
> > Additionally, we wrote another section titled “Reproducibility” in Appendix to enhance the reproducibility.
> >
> > *“A.2 Reproducibility*
> >
> > *We add more details of MaskVLM for reproducibility. We used the ImageNet pre-trained ViT (vit_base_patch_16_224) from (Wightman, 2019) and the pre-trained RoBERTa (roberta-base) from Hugging Face (Wolf et al., 2020). The detailed model architectures are visualized in Figure 7. Following (Dosovitskiy et al., 2020), the image encoder uses layer normalization (Ba et al., 2016) before each multi-head attention block while the text encoder applies layer normalization after each multi-head attention block (post norm). For the image (text) cross-modality encoder, we adopt the post norm and use the outputs of the text (image) encoder as keys and values to compute cross-attention. To compute MIM and MLM, the self-attention outputs of the masked image features, $v_m$​, is used as queries and the unmasked text features, w, are used as keys and values in the image cross-modality encoder. For the text cross-modality encoder, the masked text features, $w_m$, are used as queries and the unmasked image features, $v$, are used as keys and values. To keep the framework simple, we do not use any loss weighting for each loss term and layer decay during finetuning.”*

---

> > > ### Comment · Reviewer_cmqf · 2022-11-24
> > > **Regarding the policy**
> > >
> > > My reading of the policy differs from yours. And please note that the policy has been updated and now states 5th of June as the cutoff. The policy states:
> > >
> > > "if a paper was published (i.e., at a peer-reviewed venue) on or after June 5, 2021, authors are not required to compare their own work to that paper."
> > >
> > > In my reading, this does NOT apply to e.g. https://arxiv.org/abs/2205.14204 since it was published on arXiv on 27 May 2022.
> > >
> > > The policy DOES state that the authors "may be excused for not knowing about papers not published in peer-reviewed conference proceedings or journals, which includes papers exclusively available on arXiv.".
> > >
> > > I find it excusable that the authors did not cite the paper in their initial submission. But as per the policy I can judge the paper's main idea as not being novel since a similar idea was proposed earlier.
> > >
> > > If novelty is only required vis-a-vis conference published papers, anyone could submit a paper reinventing layernorm and saying that the idea is novel since the original layernorm paper was never published in a conference (AFAIK). Clearly this is absurd. I think that any arXiv versions which was published before the 5th of June is fair to compare against. If the paper is published at a conference after the 5th of June, I don't think the authors need to compare against the late conference version of the paper.

---

> > > > ### Author Response · Authors · 2022-12-07
> > > > **RE: Regarding the Policy**
> > > >
> > > > We thank the reviewer for the assessment of our paper, which is positive other than “missing error bars,” which we address in the next comment (thanks for the suggestion), and relation to recent work; specifically, “multiple missing references” and “novelty of the proposed method in light of these”.  The reviewer cites three specific references:
> > > >
> > > > 1. https://arxiv.org/abs/2208.10442 (BEiT-3) is a technical report posted on ArXiv on 8/22/22 that has not been accepted for publication at a peer-reviewed venue;
> > > > 2. https://arxiv.org/abs/2205.14204  (M3AE) also an Arxiv technical report posted on 5/27/22 that has never been successfully peer-reviewed, and appears to be concurrently under review at this very conference (https://openreview.net/forum?id=Z-aIURmBbBk);
> > > > 3. https://arxiv.org/abs/2204.01678 (MultiMAE) was first posted on arXiv on 4/4/22 and later published at ECCV 2022 on 10/22/22 (authors were notified of acceptance on 7/3/22).
> > > >
> > > > With regards to 1. and 2., it is *not “fair to judge the paper's main idea as not being novel since a similar idea was proposed earlier”* because there is no way to know who proposed it earlier. For example, our paper was completed prior to the dates in question, which we can document under confidentiality with the PCs, but that other contemporaneous authors could not know, unless they saw a preview, which of course we do not know, and so on and so forth.
> > > >
> > > > This is why only papers that have cleared peer-review are considered prior art, not the fact that someone posted them on ArXiv prior to being reviewed. The ICLR policy (https://iclr.cc/Conferences/2023/ReviewerGuide) states it explicitly: *“published (i.e., at a _peer-reviewed_ venue)”*. Some conferences are even more explicit and actively discourage referencing ArXiv papers. For instance, https://cvpr.thecvf.com/Conferences/2023/AuthorGuidelines states: *“A publication [...] is defined to be a written work [...] that was submitted for review by peers [...] and, after review, was accepted.” And “Note that a technical report (departmental, arXiv, etc.) version of the submission that is put up without any form of direct peer-review is NOT considered prior art and should NOT be cited”.*
> > > >
> > > > As for 3, we note that the ICLR deadline of 9/28/22 was antecedent the official publication date of ECCV, and although authors were notified earlier, the ICLR policy (https://iclr.cc/Conferences/2023/ReviewerGuide) explicitly states that “We consider papers contemporaneous if they are [...] *published* [...] on or after May 28, 2022.” So, even if we consider the author notification date instead of the actual publication date for ECCV, then 3 is contemporaneous, not antecedent, therefore not prior art.
> > > >
> > > > *Therefore, what the reviewer cites as “the biggest issue” that leads to the negative review —  in reality the only issue other than missing error bars that are addressed next —  is not a failure to reference and compare, but rather just established scholarly practice.* The conclusion that “If novelty is only required vis-a-vis conference published papers, anyone could submit a paper reinventing layernorm and saying that the idea is novel since the original layernorm paper was never published in a conference (AFAIK)” is not “absurd,” but rather consistent with sound policy (https://en.wikipedia.org/wiki/Hard_cases_make_bad_law.)
> > > >
> > > > Having said that, we completely agree with the reviewer that, given the fast pace of the field, comparing with contemporaneous work is paramount, even if it is not required by policy and not part of customary scholarly practice, if at all possible. In this specific case, considering the complexity of the models and the absence of publicly available implementations, the burden of comparison would require re-implementation of the cited methods, which is a major endeavor in its own right. Nonetheless, it is an important endeavor, that is worth exploring once all contemporaneous contributions have been vetted on their own merit, and their originality tested against prior art.

---

> > > > > ### Author Response · Authors · 2022-12-07
> > > > > **Follow up on Q2: Error bars on finetuning tasks**
> > > > >
> > > > > As suggested by the reviewer, to show that our results are not statistical outliers, we perform finetuning on COCO and Flickr30k for image-text retrieval with 5 different random seeds and report the average of Recall@k with standard deviation over all five random seeds:
> > > > >
> > > > > | Method  |   COCO  IR R@1   |   COCO  IR R@5   |   COCO IR R@10   |   COCO  TR R@1   |   COCO  TR R@5   |   COCO  TR R@10  | Flickr30k IR R@1 | Flickr30k IR R@5 | Flickr30k IR R@10 | Flickr30k TR R@1 | Flickr30k TR R@5 | Flickr30k TR R@10 |
> > > > > |:-------:|:----------------:|:----------------:|:----------------:|:----------------:|:----------------:|:----------------:|:----------------:|:----------------:|:-----------------:|:----------------:|:----------------:|:-----------------:|
> > > > > | MaskVLM | 60.05 $\pm$ 0.13 | 83.75 $\pm$ 0.08 | 90.35 $\pm$ 0.06 | 76.29 $\pm$ 0.08 | 93.71 $\pm$ 0.11 | 96.94 $\pm$ 0.09 | 84.40 $\pm$ 0.21 | 96.74 $\pm$ 0.16 |  98.32 $\pm$ 0.11 | 95.28 $\pm$ 0.23 | 99.54 $\pm$ 0.09 |  99.86 $\pm$ 0.05 |
> > > > >
> > > > >
> > > > >
> > > > > <Finetuned image-text retrieval results on COCO and Flickr30k with different random seeds. (IR: Image Retrieval, TR: Text Retrieval)>
> > > > >
> > > > > As shown in the table, the standard deviation in each R@k is marginal. It is only 0.13 and 0.08 at COCO IR R@1 and COCO TR R@1, respectively. In addition, the average R@1 for image and text retrieval on both COCO and Flickr30k are higher than the reported results of all other compared state-of-the-art methods in Table 1 of the paper (Except for ALIGN trained with 1.8 B image-text pairs). This shows that our results are not statistical outliers and MaskVLM outperforms all other compared state-of-the-art methods.

---

### Official Review · Reviewer_RfUs · 2022-10-22

**Confidence:** 3
**Correctness:** 3
**Technical Novelty And Significance:** 2
**Empirical Novelty And Significance:** 3
**Recommendation:** 6

**Clarity, Quality, Novelty And Reproducibility:**

Overall, this paper is well-written and easy to follow. However, there are some points to be further clarified.
* Figure 1 and its corresponding desciption in Introduction might lead to mislead. The proposed method relies on the random masking approach not considering explicit sematic prior. However, the examples look strongly-aligned masking. Therefore, the author need to clarify the description in Introduction.
* The author argued that "This will potentially lead to biased performance in cross-modal retrieval tasks such as image-to-text or text-to-image retrieval." This argument need reference or prelimnary empirical analysis.
* The author need to describe what ViT model is used for reproducibility. Also, I recommend writing reproducibilty section.
* For novelty, please see the weakness.


**Strength And Weaknesses:**

### Strength
* Both modality masking idea is simple and seems effective even if the idea is not very novel.
* Experimental results seem promising with significant margins compared to the baseline methods.
* Under limited pretraining data, this method show promising and competitive performances. This is meaninful for academic or small-scale industry research groups.
* Overall, the paper is clear and easy to read.

### Weakness
* [Major] Even if the main idea is clear and simple, most of them are from other previous work such as MLM, MAE, and MIM. Of course, the novelty is not all. However, for technical contribution, the combination or integration of the previous ideas needs to be not trivial and challenging, considering the nature of ICLR. Also, the authors need to describe how to effectively integrate in details. Unfortunately, I could not find the details of their proposed method. For example, there is no detailed implementation on how to make joint integration for image decoder. Figure 2 did not show the details. Figure 3 has no architecture of image cross-modal decoder. If the page limitation is an issue, the author can use the appendix.
* [Major] Some important related previous works are missed such as CoCa [Yu et al. 2022], SimVLM [Wang et al. 2022a], Florence [Yuan et al. 2021], and BEiT-3 [Wang et al. 2022b]. Among them, the author need to compare their method to SimVLM with discussion even if the pretraining dataset is not same. For the rest, the author need to introduce them in related work at least.
* [Major] For limited dataset experiments, how is the pattern on more data? I wonder the performance of MaskVLM on larger data including CC12M. For example, ALBEF presented two versions such as 4M and 14M. I think the contribution of MaskVLM can be enhanced with the same setting of ALBEF-14M. That is, it will be meaninful under 14M setup for Figure 4.
* [Minor] How is the performance on image recognition of this method, such as ImageNet-1k fine-tuning? Of course, image recongnition performance is out of scope of this paper but the comparable performance on image recongnition can improve the contribution of this paper.
*  [Minor] If I correctly understand, the training data are doubled due to the proposed mask approach. How is the performance on the same computing cost?

### References
* [Yu et al. 2022] [CoCa: Contrastive Captioners are Image-Text Foundation Models](https://arxiv.org/abs/2205.01917). arXiv:2205.01917
* [Wang et al. 2022a] [SimVLM: Simple Visual Language Model Pretraining with Weak Supervision](https://arxiv.org/abs/2108.10904). ICLR 2022.
* [Yuan et al. 2021] [Florence: A New Foundation Model for Computer Vision](https://arxiv.org/abs/2111.11432). arXiv:2111.11432.
* [Wang et al. 2022b] [Image as a Foreign Language: BEiT Pretraining for All Vision and Vision-Language Tasks](https://arxiv.org/abs/2208.10442). arXiv:2208.10442.




**Summary Of The Paper:**

This paper proposes a simple multimodal vision and language mask model for multimodal training, called MaskedVLM. The main idea is to leverage both masked image patches with full text description and masked language tokens with full image patches for semantic align.
Additionally, the method employs two pretraining losses: image-text matching and CLIP-style image-text contrastive losses.
After pretraining with 4M data, they evaluated the proposed MaskVLM on 4 standard multimodal downstream tasks: I2T, T2I, VQA, andMultimodal NLI. Experimental results look promising compared to other baseline methods and include ablation studies.

**Summary Of The Review:**

Overall, this paper proposes a simple but effective method and is easy to read. On the other hands, this paper has limited novelty and missed some important related work. If the author can address my concerns, I am willing to raise my score. I look forward to the author response.

---

> ### Author Response · Authors · 2022-11-19
> **Reply to Reviewer RfUs (1)**
>
> We are grateful for the helpful comments from the reviewer. Please find our answers below. In the paper, the updated parts are also highlighted in blue color.
>
> \
> **Q1: I could not find the details of their proposed method. For example, there is no detailed implementation on how to make joint integration for image decoder. Figure 2 did not show the details. Figure 3 has no architecture of image cross-modal decoder. If the page limitation is an issue, the author can use the appendix.**
>
> A: We added a section titled “Reproducibility” in the Appendix to explain more details of MaskVLM. Also, please refer to Figure 7 in the paper for detailed model architectures.
>
> *“A.2 Reproducibility*
>
> *We add more details of MaskVLM for reproducibility. We used the ImageNet pre-trained ViT (vit_base_patch_16_224) from (Wightman, 2019) and the pre-trained RoBERTa (roberta-base) from Hugging Face (Wolf et al., 2020). The detailed model architectures are visualized in Figure 7. Following (Dosovitskiy et al., 2020), the image encoder uses layer normalization (Ba et al., 2016) before each multi-head attention block while the text encoder applies layer normalization after each multi-head attention block (post norm). For the image (text) cross-modality encoder, we adopt the post norm and use the outputs of the text (image) encoder as keys and values to compute cross-attention. To compute MIM and MLM, the self-attention outputs of the masked image features, $v_m$​, is used as queries and the unmasked text features, w, are used as keys and values in the image cross-modality encoder. For the text cross-modality encoder, the masked text features, $w_m$​, are used as queries and the unmasked image features, $v$, are used as keys and values. To keep the framework simple, we do not use any loss weighting for each loss term and layer decay during finetuning.”*
>
> \
> **Q2: Some important related previous works are missed such as CoCa [Yu et al. 2022], SimVLM [Wang et al. 2022a], Florence [Yuan et al. 2021], and BEiT-3 [Wang et al. 2022b]. Among them, the author need to compare their method to SimVLM with discussion even if the pretraining dataset is not same. For the rest, the author need to introduce them in related work at least.**
>
> A:
>
> * Missing citations of the recent works
>
> We would like to note that according to ICLR 2023 review guidelines, our submission should not be penalized by not citing papers which are published after May 28, 2022 or not published in peer-reviewed venues (e.g. arXiv). The mentioned related works, Florence, CoCA, BEiT-3 are only available on arXiv. SimVLM is the only peer-reviewed paper. However, the authors of SimVLM did not release codes nor data (1.8 billion image-text pairs which is more than 340 times larger than what we used (5.2 million)), which makes the direct comparison impossible. Below, we quoted the review guideline from the official ICLR 2023 website for the reviewer’s convenience.
>
>
> >*Are authors expected to cite and compare with very recent work? What about non peer-reviewed (e.g., ArXiv) papers? (updated on 7 November 2022)*
> >
> >*“We consider papers contemporaneous if they are published (available in online proceedings) within the last four months. That means, since our full paper deadline is September 28, if a paper was published (i.e., at a peer-reviewed venue) on or after May 28, 2022, authors are not required to compare their own work to that paper. Authors are encouraged to cite and discuss all relevant papers, but they may be excused for not knowing about papers not published in peer-reviewed conference proceedings or journals, which includes papers exclusively available on arXiv....”*
>
> * Comparison with the recent works
>
> Although, per the review policy, it is not mandatory to cite most of the mentioned papers, we would like to discuss them in the revision, and we believe MaskVLM has sufficient differences in comparison with the related recent and concurrent works (Florence, CoCa, BEiT-3, SimVLM).
>
> **First**, no masked image modeling (MIM) is explored in Florence, CoCa, and SimVLM. Florence utilizes image-text contrastive loss only, CoCa and SimVLM focus only on language modeling without MIM. On the other hand, MaskVLM is pre-trained for both masked language modeling (MLM) and MIM.
>
> **Second**, MaskVLM is not dependent on any form of pre-trained image tokenizers but BEiT-3 is. BEiT-3 performs MIM by predicting image tokens for the masked patches. Since image tokens need to be obtained, BEiT-3 utilizes an image tokenizer trained by distilling knowledge from the CLIP model trained with 400 million image-text pairs. However, MaskVLM performs MIM by predicting masked pixels directly and does not require the image tokenizer trained with additional data.

---

> > ### Author Response · Authors · 2022-11-19
> > **Reply to Reviewer RfUs (2)**
> >
> > As suggested by the reviewer, we added the results of the SimVLM base model in Table 3 and updated the paper with discussion.
> >
> > *“Except for SimVLM whose pre-training data is more than 300 times larger than that of MaskVLM, we consistently achieve the best performances in all these tasks except for the validation split of NLVR2. In particular, MaskVLM is better than the second best method by 0.43, 1.14, and 0.27 on the test splits of VQA, NLVR2, and SNLI-VE, respectively. Compared to the base model of SimVLM, we narrow the accuracy gaps to 2.74% and 3.48% in test-std and test splits of VQA and VE, respectively. MaskVLM achieves higher accuracy than $SimVLM_{base}$​ in the test split of NLVR2 by 0.21%.”*
> >
> > The Related Work section has been also updated with additional papers.
> >
> > [Related Works]
> >
> > *“On the contrary, the authors in (Jia et al., 2021; Radford et al., 2021; Mokady et al., 2021; Shen et al., 2021; Yuan et al., 2021) show that contrastive learning with uni-modal encoders and millions of image-text pairs can achieve powerful zero-shot performance in diverse V+L tasks.”*
> >
> > *“Several V+L models focus only on predicting future text tokens without MIM (Wang et al., 2021; Yu et al., 2022; Alayrac et al., 2022). In (Bachmann et al., 2022), multiple vision modality data are masked and reconstructed to learn visual representations. While both MIM and MLM are explored in (Geng et al., 2022), the trained model is evaluated only on vision tasks. In (Dou et al., 2022; Fu et al., 2021; Singh et al., 2022; Wang et al., 2022), image tokens defined by image tokenizers trained with additional 250 million images (Ramesh et al., 2021) or distillation from the CLIP model (Radford et al., 2021) trained with 400 million image-text pairs (Peng et al., 2022) are reconstructed. In our work, we eliminate these model and data dependencies, by directly recovering RGB pixels and text tokens from masked image patches and masked text tokens. Therefore, MIM and MLM are seamlessly integrated to achieve generalizable V+L representations within a simple training framework.”*
> >
> > [References]
> >
> > *Lu Yuan, Dongdong Chen, Yi-Ling Chen, Noel Codella, Xiyang Dai, Jianfeng Gao, Houdong Hu, Xuedong Huang, Boxin Li, Chunyuan Li, et al. Florence: A new foundation model for computer vision. arXiv preprint arXiv:2111.11432, 2021.*
> >
> > *Zirui Wang, Jiahui Yu, Adams Wei Yu, Zihang Dai, Yulia Tsvetkov, and Yuan Cao. Simvlm: Simple visual language model pretraining with weak supervision. arXiv preprint arXiv:2108.10904, 2021.*
> >
> > *Jiahui Yu, Zirui Wang, Vijay Vasudevan, Legg Yeung, Mojtaba Seyedhosseini, and Yonghui Wu. Coca: Contrastive captioners are image-text foundation models. arXiv preprint arXiv:2205.01917, 2022.*
> >
> > *Wenhui Wang, Hangbo Bao, Li Dong, Johan Bjorck, Zhiliang Peng, Qiang Liu, Kriti Aggarwal, Owais Khan Mohammed, Saksham Singhal, Subhojit Som, et al. Image as a foreign language: Beit pretraining for all vision and vision-language tasks. arXiv preprint arXiv:2208.10442, 2022.*
> >
> > \
> > **Q3: For limited dataset experiments, how is the pattern on more data? I wonder the performance of MaskVLM on larger data including CC12M. For example, ALBEF presented two versions such as 4M and 14M. I think the contribution of MaskVLM can be enhanced with the same setting of ALBEF-14M. That is, it will be meaninful under 14M setup for Figure 4.**
> >
> > A: Training MaskVLM on the 4M dataset for 30 epochs using 8 V100 GPUs takes about 6 days and training on 4M + Conceptual Captions 12M can take more than two weeks . Because of this long training cycle and limited resources, we did not explore training on datasets larger than the 4M dataset, which is the most common setting in the literature of non-foundation models (e.g. ALBEF, OSCAR, UNITER), yet. We launched a training job on the 14M data but it is not finished yet due to the long training cycle. We will try our best to get the results during the discussion period but we cannot guarantee the best performances because probably the hyperparameters need to be tuned for this larger-scale experiment.

---

> > > ### Author Response · Authors · 2022-11-19
> > > **Reply to Reviewer RfUs (3)**
> > >
> > > **Q4. How is the performance on image recognition of this method, such as ImageNet-1k fine-tuning? Of course, image recongnition performance is out of scope of this paper but the comparable performance on image recongnition can improve the contribution of this paper.**
> > >
> > > A: Thanks for the great suggestion!
> > >
> > > Following CLIP (Radford et al., 2021), we perform image classification directly using the pre-trained MaskVLM on various image recognition datasets including UC Merced Land Use (Yang & Newsam, 2010), MIT-67 (Quattoni & Torralba, 2009), CUB-200 (Wah et al., 2011), Oxford Flowers (Nilsback & Zisserman, 2008), Caltech-256 (Griffin et al., 2007), and ImageNet-1K (Deng et al., 2009). We compare the Top-1 accuracy of MaskVLM with ALBEF (Li et al., 2021) in the table below. Both models are pre-trained with the 4M dataset. Since during pre-training stage, both MaskVLM and ALBEF were initialized with the ImageNet pre-trained weights, the evaluation on ImageNet is not strictly zero-shot but the evaluation on other datasets is. We formulate image classification as image-to-text retrieval where the similarity scores between a query image and all the class names are computed to retrieve top-1 class name. The similarity scores can be obtained using either ITC and ITM scores, and we report them separately. Also, the results of using prompt engineering as in CLIP are reported.
> > >
> > > |     Method    | Prompt Engineering | UC Merced Land Use | MIT-67 | CUB-200 | Oxford Flowers | Caltech256 | ImageNet-1K | Average |
> > > |:-------------:|:------------------:|:------------------:|:------:|:-------:|:--------------:|:----------:|:-----------:|:-------:|
> > > |  ALBEF (ITC)  |         No         |        31.62       |  52.46 |   5.78  |      26.93     |    45.75   |    36.28    |  33.14  |
> > > |  ALBEF (ITM)  |         No         |        26.29       |  56.19 |   5.68  |      21.68     |    41.79   |    35.41    |  31.17  |
> > > |  ALBEF (ITC)  |         Yes        |        28.38       |  51.04 |   5.33  |      25.44     |    48.71   |    30.26    |  31.53  |
> > > |  ALBEF (ITM)  |         Yes        |        15.81       |  29.70 |   3.90  |      18.86     |    26.57   |    12.53    |  17.90  |
> > > | MaskVLM (ITC) |         No         |        41.52       |  55.60 |   4.97  |      26.61     |    60.68   |    33.16    |  37.09  |
> > > | MaskVLM (ITM) |         No         |        29.52       |  58.21 |  10.36  |      36.14     |    58.95   |    34.59    |  37.96  |
> > > | MaskVLM (ITC) |         Yes        |        45.14       |  66.27 |   4.37  |      32.72     |    60.95   |    39.16    |  41.43  |
> > > | MaskVLM (ITM) |         Yes        |        40.76       |  63.36 |   8.53  |      44.53     |    61.77   |    39.71    |  43.11  |
> > >
> > > <Top-1 accuracy of pre-trained MaskVLM and ALBEF on image recognition. ITC and ITM denote the alignment scores utilized to perform image classification.>
> > >
> > > \
> > > As shown in the table, MaskVLM consistently outperforms ALBEF across all the datasets. In particular, prompt engineering improves the average accuracy over all the datasets for MaskVLM but ALBEF achieves lower average accuracy with prompt engineering. This shows that MaskVLM can better align images with variants of text than ALBEF, which results in stronger V+L representations of MaskVLM for the image recognition task.
> > >
> > > We added these evaluation results to “Appendix Section A. 5 Evaluation on image recognition”.
> > >
> > > \
> > > **Q5. If I correctly understand, the training data are doubled due to the proposed mask approach. How is the performance on the same computing cost?**
> > >
> > > A: The computational cost does not increase much by adding more tasks since most of computations are shared between tasks in MaskVLM. For example, the output of the image encoder from an unmasked image is used for the computation of ITC, ITM, and MLM. In comparison with existing works, vision-and-language (V+L) models relying on region features (e.g. UNITER, OSCAR, Uncoder-VL, 12-in-1, LXMERT) require a trained object detector to pre-extract tens or hundreds of bounding boxes per image. MaskVLM does not rely on the object detector, which significantly reduce the computational overhead. Another state-of-the-art method, ALBEF, does not have MIM loss but it requires to keep the momentum version of the model by taking the moving average of its entire parameters. Also, ITC and MLM losses in ALBEF contain additional momentum distillation loss terms. In MaskVLM, we do not keep the momentum models nor the large queues to store most recent image and text features as in ALBEF.
> > >
> > > To compare run-time of MaskVLM and ALBEF, we train both models using same machine and same batch size on the CC50% + COCO dataset for 30 epochs. The training takes about 27.5 hours for MaskVLM and 36.5 hours for ALBEF. While the run-time for MaskVLM is shorter than that for ALBEF, MaskVLM still achieves superior performance than ALBEF as shown in Table 1 ~ 4 in the paper.

---

> > > > ### Author Response · Authors · 2022-11-19
> > > > **Reply to Reviewer RfUs (4)**
> > > >
> > > > **Q6. Figure 1 and its corresponding desciption in Introduction might lead to mislead. The proposed method relies on the random masking approach not considering explicit sematic prior. However, the examples look strongly-aligned masking. Therefore, the author need to clarify the description in Introduction.**
> > > >
> > > > A: Thanks for pointing it out. While random masking is used, we picked the example that can help readers better understand the advantages of joint masked vision and language modeling. To avoid the confusion, we updated the description in Introduction by mentioning the utilization of random masking.
> > > >
> > > > *“As illustrated in Figure 1 (right part), although we exploit random masking, the dog face in the image can be used to predict the masked text token “dog” and the text “green ball” can be used to reconstruct the corresponding masked patches in the image.”*
> > > >
> > > > \
> > > > **Q7: The author argued that "This will potentially lead to biased performance in cross-modal retrieval tasks such as image-to-text or text-to-image retrieval." This argument need reference or prelimnary empirical analysis.**
> > > >
> > > > A: We empirically support this claim by comparing the image and the text retrieval performance of MaskVLM and ALBEF which is trained with ITC + ITM + MLM. In particular, we hypothesize that the significant improvement of MaskVLM over ALBEF in image retrieval is obtained by additional MIM in MaskVLM.
> > > >
> > > > To support our argument more clearly, we updated the sentences in Introduction and Section 4.3 as follows:
> > > >
> > > > [Introduction]
> > > >
> > > > *“This will potentially lead to biased performance in cross-modal retrieval tasks such as image-to-text or text-to-image retrieval as shown in our experiments.”*
> > > >
> > > > [Section 4.3]
> > > >
> > > > *“MaskVLM achieves a significant improvement over the second best method, ALBEF (Li et al., 2021), by 6.8 points at R@1 for image retrieval. Given that ALBEF is not trained for MIM, we hypothesize that ALBEF achieves the biased performance for text retrieval and MaskVLM achieves the significant improvement in image retrieval by additional MIM which models p(I∣T).”*
> > > >
> > > > \
> > > > **Q8: The author need to describe what ViT model is used for reproducibility. Also, I recommend writing reproducibilty section.**
> > > >
> > > > A: As answered in Q1, the section “Reproducibility” has been added in the Appendix with the detailed model architectures.

---

> > > > > ### Comment · Reviewer_RfUs · 2022-11-24
> > > > > **Thank you for your response**
> > > > >
> > > > > I thank the authors for their efforts to alleviate my concerns. Most of my concerns were addressed by the authors' responses. For citation, I also already read the reviewer's guidelines. I would like to note the reason why I recommend citing recent work such as Florence and CoCa is that it can make the paper's quality higher. Of course, experimental comparisons are not mandatory.  Anyway, I raised my score considering the novelty.

---

> > > > > > ### Author Response · Authors · 2022-12-07
> > > > > > **Thank you**
> > > > > >
> > > > > > We are glad to hear that our responses addressed your concerns and thank you for raising the score considering the novelty of the paper. We also believe that the added discussion on the recent papers made the quality of this paper higher.

---

### Official Review · Reviewer_kwqv · 2022-10-24

**Confidence:** 4
**Correctness:** 2
**Technical Novelty And Significance:** 2
**Empirical Novelty And Significance:** 2
**Recommendation:** 6

**Clarity, Quality, Novelty And Reproducibility:**

Clarity: For the most part clear. The method exposition and the math could be tidied up a bit.

Quality: Decent quality.

Novelty: Somewhat novel. MaskVLM is a rearrangement of existing basic building blocks (cross-attention, ITM loss, contrastive loss, MLM/MIM) from existing papers in a novel way.

Reproducibility: Could probably be roughly reimplemented from the details in the paper and the appendix, although a few details are missing (pre or post norm in the transformers, layer decay during fine tuning, weighting on the three losses, etc).

**Strength And Weaknesses:**

Strengths:

The authors compare against a large number of baselines, and are able to show better performance in almost all instances.

Qualitative results as well as ablation of the objectives for pre-training provide compelling evidence that cross-modal masked modeling can help to produce better representations.

Well designed experiment shows that in the low-data regime, adding a masked image modeling loss can improve performance by making more efficient use of the data.

Good ablation studies of masking ratios and strategies in the appendix.

Weaknesses:

One significant problem with the paper is that many recent and concurrent methods and references are missing. As a result, many of the tables that claim to compare MaskVLM to SOTA results do not actually:

- Table 1 lists SOTA methods on MSCOCO and Flickr30k image-text retrieval with finetuning. It is however missing Florence, which beats all methods in the table.
- Table 2 lists SOTA methods on zero-shot Flickr30k image-text retrieval. It is also missing Florence, and CoCA.
- Table 3 lists SOTA methods on VQA but is missing the above methods and SimVLM and Flamingo.

If the claim is that MaskVLM achieves SOTA with specific qualifications (such as pre-training on a smaller dataset) this should be stated and shown explicitly. If the claim is that MaskVLM achieves SOTA outright, then I don't think this is true.

Some papers that should be cited:
- Florence
- CoCA and SimVLM
- Flamingo
- MultiMAE
- M3AE (could be optional I think)
- BEiT-3 (this came out after ICLR submission deadline, but is relevant)

In addition, claims such as "In the domain of V+L learning, there exist limited works on end-to-end joint modeling of V+L signals" are untrue, in light of the above references.

The method and its presentation are also a tad complicated. It could help significantly if the explanation of the method was cleaned up a bit. Here are some things that I found confusing:

- The first term in Eq. 1 appears to show the MLM loss is applied to unmasked tokens as well as masked tokens, as opposed to just masked tokens as in BERT. Is this a typo, or actually the case? And if so, why was this decision made?
- In Eq. 1, there is an expectation over D on the second term but not the first. I'm assuming this is a typo?
- One more nitpick for eq 1: I think $ g_{im}^{de} $ in the definition of $ \phi_{im} $ should also take $ f_{txt}(T) $ as an argument, as the image decoder is also allowed to cross-attend to vision text information.
- Why use an L1 loss for MIM rather than L2 like in MAE?
- In order to fuse representations, element-wise multiplication is used (in the downstream task heads, and the ITM loss). This seems non-standard to me, so perhaps a reference to previous work that does this could be helpful.

One small nitpick: MAE is cited as a preprint, but it was published at CVPR 2022.

**Summary Of The Paper:**

The authors propose masked language and image modeling to pre-train image and text representations for downstream tasks such as image-text retrieval and various image-language reasoning tasks (VQA, NLVR, visual entailment). The proposed model is trained on paired image-text data, where one modality is masked while the other is kept unmasked. Given these two inputs the goal is to reconstruct the masked modality. In addition, a contrastive text-image loss (like CLIP) and an image-text matching loss are used.

The authors compare against a multitude of baselines on the tasks, which for the wide majority they beat. In addition, they show that their method works better in the low-data regime, probably because of the additional masked image modeling objective. The losses in the model are then ablated on the image-text retrieval text and it is shown that adding MIM and MLM improves the model. Finally, qualitative results are presented showing how additional cross-modal information helps prediction. Ablations on masking strategy and ratios are given in the appendix.

**Summary Of The Review:**

This paper presents a somewhat novel method for image-language pre-training and shows that it beats a large number of baselines. The problem is that it omits quite a few recent or concurrent baselines in the experimental and related work section. The claims of SOTA in the paper then seem somewhat dubious to me. See the weaknesses section for more details.

---

> ### Author Response · Authors · 2022-11-19
> **Reply to Reviewer kwqv (1)**
>
> We appreciate the constructive feedback from the reviewer. Please find our answers to the questions below. In the paper, the updated parts are also highlighted in blue color.
>
> \
> **Q1: Many recent and concurrent methods and references are missing.**
>
> A:
>
> * Missing citations of the recent works
>
> We would like to note that according to ICLR 2023 review guidelines, our submission should not be penalized by not citing papers which are published after May 28, 2022 or not published in peer-reviewed venues (e.g. arXiv). The mentioned related works, **Florence, CoCA, Flamingo, BEiT-3, M3AE, are only available on arXiv. Also, M3AE is the concurrent submission to ICLR 2023. MutilMAE was published at ECCV 2022 where the final acceptance decision was notified on July 3, 2022 which is after May 28, 2022.** Among the suggested related works, SimVLM is the only peer-reviewed paper. However, the authors of SimVLM did not release codes nor data (1.8 billion image-text pairs which is more than 340 times larger than what we used (5.2 million)), which makes the direct comparison impossible. Below, we quoted the review guideline from the official ICLR 2023 website for the reviewer’s convenience.
>
> >*Are authors expected to cite and compare with very recent work? What about non peer-reviewed (e.g., ArXiv) papers? (updated on 7 November 2022)*
> >
> >*“We consider papers contemporaneous if they are published (available in online proceedings) within the last four months. That means, since our full paper deadline is September 28, if a paper was published (i.e., at a peer-reviewed venue) on or after May 28, 2022, authors are not required to compare their own work to that paper. Authors are encouraged to cite and discuss all relevant papers, but they may be excused for not knowing about papers not published in peer-reviewed conference proceedings or journals, which includes papers exclusively available on arXiv....”*
>
>
> * Comparison with the recent works
>
> Although, per ICLR 2023 review policy, it is not mandatory to cite most of the mentioned papers, we would like to discuss them in the revision, and we believe MaskVLM has sufficient differences in comparison with the related recent and concurrent works (Florence, CoCa, Flamingo, BEiT-3, M3AE, MultiMAE, SimVLM).
>
> **First**, no masked image modeling (MIM) is explored in Florence, CoCa, Flamingo, and SimVLM. Florence utilizes image-text contrastive loss only. Flamingo and SimVLM focus only on language modeling without MIM. On the other hand, MaskVLM is pre-trained for both masked language modeling (MLM) and MIM.
>
> **Second**, MaskVLM is not dependent on any form of pre-trained image tokenizers but BEiT-3 is. BEiT-3 performs MIM by predicting image tokens for the masked patches. Since image tokens need to be obtained, BEiT-3 utilizes an image tokenizer trained by distilling knowledge from the CLIP model trained with 400 million image-text pairs. However, MaskVLM performs MIM by predicting masked pixels directly and does not require the image tokenizer trained with additional data.
>
> **Third**, MultiMAE is not trained with the language data but with different vision modality data (e.g. RGB, depth, semantic segmentation). Also, both MultiMAE and M3AE are not evaluated on V+L downstream tasks but only on vision downstream tasks. MaskVLM is trained to align V+L data and also evaluated on diverse V+L downstream tasks.
>
> To highlight these differences, we first added the results of the SimVLM base model in Table 3 and updated the paper with discussion as follows:
>
> *“Except for SimVLM whose pre-training data is more than 300 times larger than that of MaskVLM, we consistently achieve the best performances in all these tasks except for the validation split of NLVR2. In particular, MaskVLM is better than the second best method by 0.43, 1.14, and 0.27 on the test splits of VQA, NLVR2, and SNLI-VE, respectively. Compared to the base model of SimVLM, we narrow the accuracy gaps to 2.74% and 3.48% in test-std and test splits of VQA and VE, respectively. MaskVLM achieves higher accuracy than $SimVLM_{base}$​ in the test split of NLVR2 by 0.21%.”*
>
> Also, the Related Work section has been updated with additional papers as follows:
>
> [Related Work]
>
> *“On the contrary, the authors in (Jia et al., 2021; Radford et al., 2021; Mokady et al., 2021; Shen et al., 2021; Yuan et al., 2021) show that contrastive learning with uni-modal encoders and millions of image-text pairs can achieve powerful zero-shot performance in diverse V+L tasks.”*
>
> *“In (Bachmann et al., 2022), multiple vision modality data are masked and reconstructed to learn visual representations.”*

---

> > ### Author Response · Authors · 2022-11-19
> > **Reply to Reviewer kwqv (2)**
> >
> > *“Several V+L models focus only on predicting future text tokens without MIM (Wang et al., 2021; Yu et al., 2022; Alayrac et al., 2022). While both MIM and MLM are explored in (Geng et al., 2022), the trained model is evaluated only on vision tasks. In (Dou et al., 2022; Fu et al., 2021; Singh et al., 2022; Wang et al., 2022), image tokens defined by image tokenizers trained with additional 250 million images (Ramesh et al., 2021) or distillation from the CLIP model (Radford et al., 2021) trained with 400 million image-text pairs (Peng et al., 2022) are reconstructed. In our work, we eliminate these model and data dependencies, by directly recovering RGB pixels and text tokens from masked image patches and masked text tokens. Therefore, MIM and MLM are seamlessly integrated to achieve generalizable V+L representations within a simple training framework.”*
> >
> > [References]
> >
> > *Lu Yuan, Dongdong Chen, Yi-Ling Chen, Noel Codella, Xiyang Dai, Jianfeng Gao, Houdong Hu, Xuedong Huang, Boxin Li, Chunyuan Li, et al. Florence: A new foundation model for computer vision. arXiv preprint arXiv:2111.11432, 2021.*
> >
> > *Zirui Wang, Jiahui Yu, Adams Wei Yu, Zihang Dai, Yulia Tsvetkov, and Yuan Cao. Simvlm: Simple visual language model pretraining with weak supervision. arXiv preprint arXiv:2108.10904, 2021.*
> >
> > *Jiahui Yu, Zirui Wang, Vijay Vasudevan, Legg Yeung, Mojtaba Seyedhosseini, and Yonghui Wu. Coca: Contrastive captioners are image-text foundation models. arXiv preprint arXiv:2205.01917, 2022.*
> >
> > *Jean-Baptiste Alayrac, Jeff Donahue, Pauline Luc, Antoine Miech, Iain Barr, Yana Hasson, Karel Lenc, Arthur Mensch, Katie Millican, Malcolm Reynolds, et al. Flamingo: a visual language model for few-shot learning. arXiv preprint arXiv:2204.14198, 2022.*
> >
> > *Roman Bachmann, David Mizrahi, Andrei Atanov, and Amir Zamir. Multimae: Multi-modal multi-task masked autoencoders. arXiv preprint arXiv:2204.01678, 2022.*
> >
> > *Xinyang Geng, Hao Liu, Lisa Lee, Dale Schuurams, Sergey Levine, and Pieter Abbeel. Multimodal masked autoencoders learn transferable representations. arXiv preprint arXiv:2205.14204, 2022.*
> >
> > *Wenhui Wang, Hangbo Bao, Li Dong, Johan Bjorck, Zhiliang Peng, Qiang Liu, Kriti Aggarwal, Owais Khan Mohammed, Saksham Singhal, Subhojit Som, et al. Image as a foreign language: Beit pretraining for all vision and vision-language tasks. arXiv preprint arXiv:2208.10442, 2022.*
> >
> > \
> > **Q2. If the claim is that MaskVLM achieves SOTA with specific qualifications (such as pre-training on a smaller dataset) this should be stated and shown explicitly. If the claim is that MaskVLM achieves SOTA outright, then I don't think this is true.**
> >
> > A: Thanks for the suggestion. We agree that we should clarify MaskVLM achieves SOTA results in the regime of millions of pre-training data. Therefore, we updated the abstract as follows:
> >
> > *“Our experiments on various V+L tasks show that the proposed method, along with common V+L alignment losses, achieves state-of-the-art performance in the regime of millions of pre-training data. Also, we outperforms the other competitors by a significant margin in limited data scenarios.”*
> >
> > \
> > **Q3. In addition, claims such as "In the domain of V+L learning, there exist limited works on end-to-end joint modeling of V+L signals" are untrue, in light of the above references.**
> >
> > A: We agree. We updated the sentence not to claim that there exist limited works. Instead, we added more papers related to mask V+L modeling in the Related Work section as follows:
> >
> > *“In the domain of V+L learning, (Arici et al., 2021) explores MIM and MLM for catalog data with short text attributes. V+L models with an object detector often aim at recovering only bounding box visual features (Chen et al., 2020b; Li et al., 2020a; Lu et al., 2020; Tan & Bansal, 2019; Su et al., 2019). Several V+L models focus only on predicting future text tokens without MIM (Wang et al., 2021; Yu et al., 2022; Alayrac et al., 2022). While both MIM and MLM are explored in (Geng et al., 2022), the trained model is evaluated only on vision tasks. In (Dou et al., 2022; Fu et al., 2021; Singh et al., 2022; Wang et al., 2022), image tokens defined by image tokenizers trained with additional 250 million images (Ramesh et al., 2021) or distillation from the CLIP model (Radford et al., 2021) trained with 400 million image-text pairs (Peng et al., 2022) are reconstructed.”*
> >
> > \
> > **Q4. The first term in Eq. 1 appears to show the MLM loss is applied to unmasked tokens as well as masked tokens, as opposed to just masked tokens as in BERT. Is this a typo, or actually the case? And if so, why was this decision made?**
> >
> > A: Thanks for pointing it out. We confirm that it is a typo and MLM loss is computed for masked tokens only as in BERT. We updated the paper and please refer to the answer of Q6 for the updated version.

---

> > > ### Author Response · Authors · 2022-11-19
> > > **Reply to Reviewer kwqv (3)**
> > >
> > > **Q5. In Eq. 1, there is an expectation over D on the second term but not the first. I'm assuming this is a typo?**
> > >
> > > A. This is a typo . Equation 1 and its explanation are updated. Please refer to the answer of Q6 for the updated version.
> > >
> > > \
> > > **Q6. I think $g_{im}^{de}$​ in the definition of $\phi_{im}$ should also take $f_{txt}(T)$ as an argument, as the image decoder is also allowed to cross-attend to vision text information.**
> > >
> > > A: In the current formulation, $g_{im}^{de}$​ takes $f_{txt}(T)$ as an argument. Equation 1 and its explanation are updated as follows:
> > >
> > > *"The masked V+L modeling loss, $\mathcal{L}_{MVLM}$​, is defined as*
> > >
> > > $\mathcal{L}_{MVLM} =\mathbb{E}\_{(I, T) \sim D } [\, \underbrace{\mathcal{H}(y^{M}\_T, \phi\_{txt}^{M}(I, T\_m))}\_\text{MLM} + \underbrace{\dfrac{1}{\Omega({I^M})} \|I^{M} - \phi\_{im}^{M}(I\_m, T)\|\_{1}}\_\text{MIM} ]\, $
> > >
> > > *where $\phi_{txt} = g_{txt}^{de}(g_{txt}(f_{im}(I), f_{txt}(T_{m})))$ and $\phi_{im} = g_{im}^{de}(g_{im}(f_{im}(I_{m}), f_{txt}(T)))$. The loss is computed only for masked pixels and text tokens. Hence, the superscript $M$ denotes data or features correspond to the masked signal. A pair of $I$ and $T$ is sampled from the training dataset $D$, $\mathcal{H}$ denotes cross-entropy, and $y^{M}_T$​ is a matrix that contains one-hot row vectors for the ground truth of masked text tokens.”*
> > >
> > > \
> > > **Q7. Why use an L1 loss for MIM rather than L2 like in MAE?**
> > >
> > > A: We follow SimMIM, which was also cited in the paper, to use the L1 loss for MIM.
> > >
> > > *“Zhenda Xie, Zheng Zhang, Yue Cao, Yutong Lin, Jianmin Bao, Zhuliang Yao, Qi Dai, and Han Hu. Simmim: A simple framework for masked image modeling. arXiv preprint arXiv:2111.09886, 2021.”*
> > >
> > > \
> > > **Q8. In order to fuse representations, element-wise multiplication is used (in the downstream task heads, and the ITM loss). This seems non-standard to me, so perhaps a reference to previous work that does this could be helpful.**
> > >
> > > A: Thanks for the suggestion. We follow the fusion mechanism used in ViLBERT and the paper is cited as you suggested.
> > >
> > > *“To fuse these two features, we compute the element-wise product of $z_{cross}^{im}$​ and $z_{cross}^{txt}$​ ($z_{cross}^{im} * z_{cross}^{txt}$), and a FC layer followed by softmax is applied to obtain the final prediction (Lu et al., 2019).”*
> > >
> > > [Reference]
> > >
> > > *“Jiasen Lu, Dhruv Batra, Devi Parikh, and Stefan Lee. Vilbert: Pretraining task-agnostic visiolinguistic representations for vision-and-language tasks. Advances in neural information processing systems, 32, 2019”*
> > >
> > > \
> > > **Q9. MAE is cited as a preprint, but it was published at CVPR 2022.**
> > >
> > > A: The citation for MAE has been updated as follows:
> > >
> > > *“Kaiming He, Xinlei Chen, Saining Xie, Yanghao Li, Piotr Doll ́ar, and Ross Girshick. Masked autoencoders are scalable vision learners. In Proceedings of the IEEE/CVF Conference on Computer Vision and Pattern Recognition, pp. 16000–16009, 2022.”*
> > >
> > > \
> > > **Q10. A few details are missing (pre or post norm in the transformers, layer decay during fine tuning, weighting on the three losses, etc)**
> > >
> > > A. We added a section titled “Reproducibility” in the Appendix to explain more details of MaskVLM. Also, please refer to Figure 7 in the paper for detailed model architectures.
> > >
> > > *“A.2 Reproducibility*
> > >
> > > *We add more details of MaskVLM for reproducibility. We used the ImageNet pre-trained ViT (vit_base_patch16_224) from (Wightman, 2019) and the pre-trained RoBERTa (roberta-base) from Hugging Face (Wolf et al., 2020). The detailed model architectures are visualized in Figure 7. Following (Dosovitskiy et al., 2020), the image encoder uses layer normalization (Ba et al., 2016) before each multi-head attention block while the text encoder applies layer normalization after each multi-head attention block (post norm). For the image (text) cross-modality encoder, we adopt the post norm and use the outputs of the text (image) encoder as keys and values to compute cross-attention. To compute MIM and MLM, the self-attention outputs of the masked image features, $v_m$​, is used as queries and the unmasked text features, w, are used as keys and values in the image cross-modality encoder. For the text cross-modality encoder, the masked text features, $w_m$​, are used as queries and the unmasked image features, $v$, are used as keys and values. To keep the framework simple, we do not use any loss weighting for each loss term and layer decay during finetuning.”*

---

> > > > ### Comment · Reviewer_kwqv · 2022-11-30
> > > > **Reply to Authors**
> > > >
> > > > Thank you authors for the detailed response. Most of my points have been addressed and in light of the reviewer guidelines on contemporaneous work, which the authors kindly pointed out, I think my main concern has been resolved. I appreciate the changes and edits that the authors have made and am raising my score.

---

> > > > > ### Author Response · Authors · 2022-12-07
> > > > > **Thank you**
> > > > >
> > > > > We thank the reviewer for all the detailed comments and increasing the score. We also believe that addressing your comments enhanced the quality of the paper.

---

### Official Review · Reviewer_UZ4d · 2022-10-24

**Confidence:** 4
**Correctness:** 4
**Technical Novelty And Significance:** 3
**Empirical Novelty And Significance:** 3
**Recommendation:** 8

**Clarity, Quality, Novelty And Reproducibility:**

The paper is well written and the proposed method is novel enough. The authors do provide a lot of details about the training process, so it might be possible for the results to be reproduced.

**Strength And Weaknesses:**

Strengths
The paper is well written and easy to follow.

An interesting approach based on jointly masking images and text for learning visual and text representations is presented.

Experiments on multiple tasks and datasets are presented. A detailed ablation study is also conducted.

Weaknesses.

In several occasions the authors claim that other methods have been trained on much larger datasets. As a consequence this makes comparison with these works problematic. Is it possible that these methods are trained on the same amount of data as the one used in this study? Alternatively, would it be possible that the proposed method is trained on much larger datasets?

In order to investigate the impact of the each loss term it would be desirable that only one of them is removed at a time. In other words, it would it would be very informative to also include results with the following losses (in Table 5): MLM + MIM + ITC and MLM + MIM + ITM.


**Summary Of The Paper:**

This paper proposes the use of joint masking for vision and language for learning representations from text and images. The proposed approach is tested on multiple tasks and achieves state-of-the-art results especially in the case of limited training data.

**Summary Of The Review:**

A very interesting contribution with several experiments and a detailed ablation study.

---

> ### Author Response · Authors · 2022-11-19
> **Reply to Reviewer UZ4d (1)**
>
> We would like to thank the reviewer for positive feedback. Below, we carefully answered to the reviewer’s questions.
>
> \
> **Q1. In several occasions the authors claim that other methods have been trained on much larger datasets. As a consequence this makes comparison with these works problematic. Is it possible that these methods are trained on the same amount of data as the one used in this study? Alternatively, would it be possible that the proposed method is trained on much larger datasets?**
>
> A: CLIP and ALIGN are trained with 400 million and 1.8 billion image-text pairs which are more than 76 times and 343 times of data used for MaskVLM, respectively. Also, data used for training CLIP and ALIGN are not publicly available. Since it is impossible to scale up MaskVLM at this scale given our resource constraints and the same pre-training data is not available, we train the CLIP model using the codes from the OpenCLIP (https://github.com/mlfoundations/open_clip) project, which aims at reproducing OpenAI CLIP results, with the same 4M dataset used by MaskVLM. In particular, we compare zero-shot and finetuned image-text retrieval results on Flickr30k in the table below. The CLIP model is trained with a batch size of 3092 using the same number of machines as MaskVLM. Following the original CLIP paper, both vision and text encoders are trained from scratch. A learning rate of 5e-4 and 1e-4 are used for pre-training and finetuning, respectively.
>
> |  Method | Zero-shot IR R@1 | Zero-shot IR R@5 | Zero-shot TR R@1 | Zero-shot TR R@5 | Finetuned IR R@1 | Finetuned IR R@5 | Finetuned TR R@1 | Finetuned TR R@5 |
> |:-------:|:----------------:|:----------------:|:----------------:|:----------------:|:----------------:|:----------------:|:----------------:|:----------------:|
> |   CLIP  |       30.2       |       56.8       |       43.9       |       73.1       |       47.9       |       74.6       |       64.1       |       86.2       |
> | MaskVLM |       **75.0**       |       **92.5**       |       **87.0**       |       **97.9**       |       **84.5**       |       **96.7**       |       **95.6**       |       **99.4**       |
>
> <Comparison of zero-shot image-text retrieval results between CLIP and MaskVLM on Flickr30k . (IR: Image Retrieval, TR: Text Retrieval)>
>
> Since large-scale data is the most important factor for CLIP to achieve powerful performance, the performance is significantly degraded when the CLIP is trained with this 4M dataset, and MaskVLM outperforms CLIP in the same data regime.
>
> In addition, we evaluate on zero-shot image classification, where the CLIP model is mainly evaluated on. The datasets include UC Merced Land Use, MIT-67, CUB-200, Oxford Flowers, Caltech256. As shown in the table below, when the same data is used, MaskVLM consistently outperforms CLIP on zero-shot image classification.
>
> |  Models | UC Merced Land Use | MIT-67 | CUB-200 | Oxford Flowers | Caltech256 |
> |:-------:|:------------------:|:------:|:-------:|:--------------:|:----------:|
> |   CLIP  |        20.19       |  36.72 |   6.32  |      23.83     |    44.35   |
> | MaskVLM |        **40.76**       |  **63.36** |   **8.53**  |      **44.53**     |    **61.77**   |
>
> <Comparison of zero-shot image recognition performance (Top-1 accuracy) between CLIP and MaskVLM on Flickr30k>

---

> > ### Author Response · Authors · 2022-11-19
> > **Reply to Reviewer UZ4d (2)**
> >
> > **Q2. In order to investigate the impact of the each loss term it would be desirable that only one of them is removed at a time. In other words, it would it would be very informative to also include results with the following losses (in Table 5): MLM + MIM + ITC and MLM + MIM + ITM.**
> >
> > A: In the MaskVLM framework, ITM cannot be used alone without ITC, because the hard negatives used for ITM are obtained by ranking the scores of ITC. Therefore, we cannot show the result of  MLM + MIM + ITM, but we added the results using ITC + MLM + MIM in the table below (Row 3).
> >
> > |          Loss         | Finetuned IR R@1 | Finetuned IR R@5 | Finetuned TR R@1 | Finetuned TR R@5 | Zero-shot IR R@1 | Zero-shot IR R@5 | Zero-shot TR R@1 | Zero-shot TR R@5 |
> > |:---------------------:|:----------------:|:----------------:|:----------------:|:----------------:|:----------------:|:----------------:|:----------------:|:----------------:|
> > |          ITC          |       65.10      |       89.88      |       80.10      |       96.90      |       55.08      |       80.90      |       68.40      |       90.00      |
> > |        MLM +MIM       |       76.08      |       94.40      |       90.30      |       98.80      |        N/A       |        N/A       |        N/A       |        N/A       |
> > |     ITC + MLM +MIM    |       78.06      |       94.90      |       89.70      |       99.10      |       57.02      |       84.10      |       69.70      |       91.20      |
> > |       ITC + ITM       |       79.96      |       95.56      |       92.30      |       98.90      |       69.50      |       89.54      |       82.40      |       96.60      |
> > |    ITC + ITM + MLM    |       80.34      |       95.82      |       92.00      |       99.30      |       70.74      |       90.92      |       84.40      |       97.10      |
> > |    ITC + ITM + MIM    |       80.12      |       95.56      |       91.50      |       99.00      |       69.26      |       90.30      |       82.90      |       97.20      |
> > | ITC + ITM + MLM + MIM |       **81.26**      |       **96.00**      |       **94.10**      |       **99.60**      |       **71.18**      |       **91.12**      |       **85.60**      |       **97.50**      |
> >
> > <Image-text retrieval evaluation on Flickr30k with different loss functions for pre-training. (IR: Image Retrieval, TR: Text Retrieval)>
> >
> > By comparing between ITC (Row 1) and ITC + MLM + MIM (Row 3), we show that by adding MLM + MIM significantly improves the image and text retrieval R@1 by 12.96 and 9.60, respectively when models are finetuned. Also, the zero-shot image and text retrieval R@1 is improved by 1.94 and 1.30, respectively. This shows that the MLM + MIM does help to train more powerful feature representation for vision and language tasks.

---

> > > ### Comment · Reviewer_UZ4d · 2022-11-28
> > > **Reply**
> > >
> > >
> > > I would like to thank the authors for addressing my concerns.

---

> > > > ### Author Response · Authors · 2022-12-07
> > > > **Thank you**
> > > >
> > > > We appreciate your encouraging comments and positive rating. We are glad to hear that you found the paper is well written and the proposed idea is novel enough.

---

### Decision · Program_Chairs · 2023-01-20

**Decision:**

Accept: poster

**Justification For Why Not Higher Score:**

The reviewers' main criticism is the limited novelty -- authors show that cross-modal masked language modeling leads to improvement, and how to effectively train such a model in limited (~4M) data regime. While reviewers vote for acceptance, the expected impact of the paper falls short of spotlight or oral contributions. Successful work on other domains (e.g. video-text modeling) would be a way to achieve that level of significance.


**Justification For Why Not Lower Score:**

Authors and reviewers discussed recent (or concurrent) work on the merits and authors took a positive attitude towards including other work in the final draft of the paper, which ultimately confirmed the efficacy of the proposed method, and results in a well-rounded paper. All reviewers find the work to be significant.

**Metareview: Summary, Strengths And Weaknesses:**

Summary

This paper proposes MaskedVLM, a "simple" multimodal vision and language masking training procedure. The method employs two pretraining losses: image-text matching and CLIP-style image-text contrastive loss. The main idea is to leverage both masked image patches with full text description and masked language tokens with full image patches for cross-modal reconstruction and semantic alignment. After pretraining with 4M data points, authors evaluate the proposed MaskVLM on 4 standard multimodal downstream tasks: I2T, T2I, VQA, and multimodal NLI. Experimental results show that the method outperforms competing and contemporary methods specifically in a low data regime, and authors include several insightful ablation studies.

Strengths

- Reviewers agree that the cross-modal masking idea is an interesting (if somewhat obvious) approach, which deserves to be presented at a conference. The idea may not be transformative, but it is simple and seems effective, and well described in this paper
- Experiments on multiple tasks and (open source) datasets are presented. Detailed ablation studies are also conducted, confirming that cross-modal masking produces better representations in a low data (millions of samples) regime, and the contributions of the individual losses. The approach outperforms most baselines and competing methods.
- Specifically, the idea of adding a masked image modeling loss to make more efficient use of the data should be helpful, and researchers (even with "academic" resources) should be able to reproduce these results according to the reviewers' estimation
- The paper is generally well-written, easy to follow, and after revisions, reviewers are satisfied that the paper includes a thorough and compliant discussion of the related work and state of the art

Weaknesses

- The submitted version was missing multiple recent references and a thorough discussion of related work (CoCa [Yu et al. 2022], SimVLM [Wang et al. 2022a], Florence [Yuan et al. 2021], and BEiT-3 [Wang et al. 2022b)
- Authors need to clarify the conditions under which the proposed work outperforms baselines and comparable work
- Even if the main ideas are clear and simple, most of them are from other previous work such as MLM, MAE, and MIM. Authors need to clearly describe their implementation in detail so that it can be reproduced
- In order to investigate the impact of the each loss term it would be desirable that only one of them is removed at a time. In other words, it would it would be very informative to also include results with the following losses (in Table 5): MLM + MIM + ITC and MLM + MIM + ITM.


**Note From Pc:**

if the above contains the word "oral" or "spotlight" please see: "oral" presentation means -> notable-top-5% and "spotlight" means -> notable-top-25%. As stated in our emails, we are disassociating presentation type from AC recommendations

**Summary Of Ac-Reviewer Meeting:**

n/a